evolution

major evolutionary transitions, multicellularity, scaling, body size, cell types, organizational complexity

**Author for correspondence:**
R. M. Fisher
e-mail: bertifisher@gmail.com

# The evolution of multicellular complexity: the role of relatedness and environmental constraints

R. M. Fisher[1], J. Z. Shik[1,2] and J. J. Boomsma[1]

[1]Section for Ecology and Evolution, Department of Biology, University of Copenhagen, Denmark
[2]Smithsonian Tropical Research Institute, Apartado 0843-03092, Balboa, Ancon, Republic of Panama

RMF, 0000-0003-4094-5431; JJB, 0000-0002-3598-1609

A major challenge in evolutionary biology has been to explain the variation in multicellularity across the many independently evolved multicellular lineages, from slime moulds to vertebrates. Social evolution theory has highlighted the key role of relatedness in determining multicellular complexity and obligateness; however, there is a need to extend this to a broader perspective incorporating the role of the environment. In this paper, we formally test Bonner's 1998 hypothesis that the environment is crucial in determining the course of multicellular evolution, with aggregative multicellularity evolving more frequently on land and clonal multicellularity more frequently in water. Using a combination of scaling theory and phylogenetic comparative analyses, we describe multicellular organizational complexity across 139 species spanning 14 independent transitions to multicellularity and investigate the role of the environment in determining multicellular group formation and in imposing constraints on multicellular evolution. Our results, showing that the physical environment has impacted the way in which multicellular groups form, highlight that environmental conditions might have affected the major evolutionary transition to obligate multicellularity.

## 1. Introduction

Macroscopic life on earth has been shaped by the evolution of multicellularity from unicellular ancestors. Multicellularity ranges from simple cell aggregations found in yeast to differentiated metazoan organisms, with much diversity in between [1]. For example, our bodies contain $10^{14}$ cells with more than 200 specialized types [2] but *Volvox* is 10 orders of magnitude smaller and has just two cell types [3]. Some lineages have become obligately multicellular, where cells only exist as part of a multicellular organism (e.g. animals), whereas others remain facultative, switching between a unicellular and multicellular lifestyle (e.g. cellular slime moulds) [4].

A major challenge in evolutionary biology has been to explain this variation in complexity among multicellular lineages. Social evolution theory has greatly advanced our understanding of the evolution of multicellularity, primarily through clarifying the factors that favour the cooperation needed to become multicellular. We understand how relatedness between cells is crucial in determining when altruism can evolve, as for example in *Dictyostelium* slime moulds [5] where division of labour between cell types [6] and the proliferation of cheaters [7–9] have been amply studied. It has also become clear that clonal relatedness ($r = 1$) is a necessary, albeit not sufficient, condition for the evolution of obligate multicellularity like we see in animals and plants, and that these lineages have more cell types than those with facultative multicellularity [4].

However, there are limits to the variation in multicellular complexity that is explained by relatedness. For example, both land plants and fungi have cells that are clonal and obligately multicellular, but plants have approximately

Proc. R. Soc. B 287: 20192963

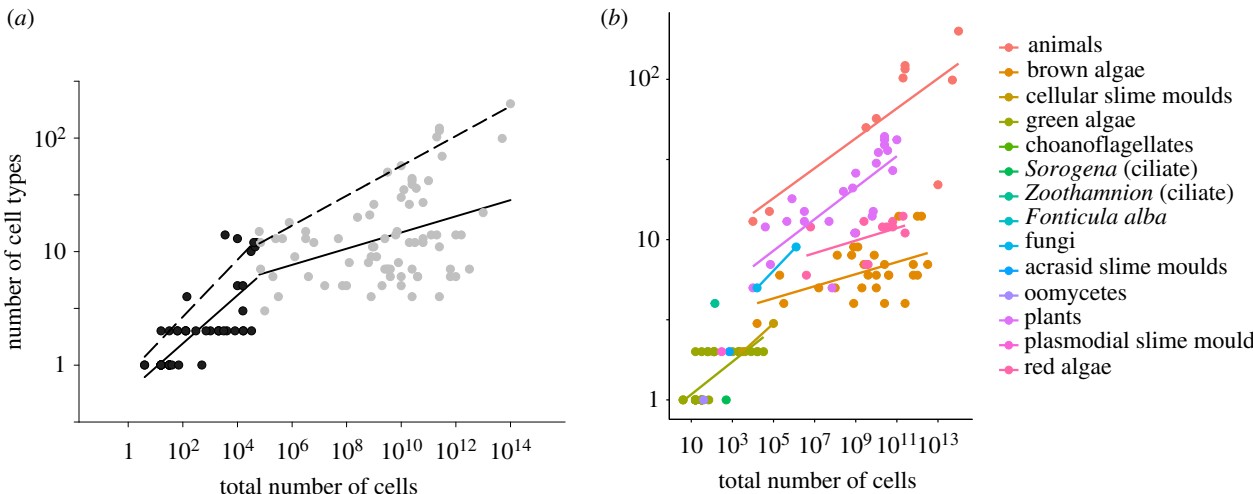

**Figure 1.** (a) Scaling across multicellular species. The relationship between number of cell types and total number of cells for smaller (in black) and larger multicellular species (in grey) shown on logarithmic axes. Small species (i.e. those below $6.3 \times 10^4$ cells) show a steeper allometry (OLS regression: slope = 0.21 (confidence interval = 0.16 − 0.26) compared to large species (i.e. those larger than $6.3 \times 10^4$ cells; OLS regression: slope = 0.07 (confidence interval = 0.03 − 0.11). Solid lines show the OLS regression and dashed lines show regressions through the upper 90% quantile of the data. We estimated the breakpoint of $4.8 \pm 0.9$ (corresponding to $6.3 \times 10^4$ total cells) using the 'segmented' package in R. (b) Multicellular organizational complexity across different multicellular lineages. Organizational complexity, measured as both the number of cell types and the total number of cells, for each of the independently evolved multicellular lineages. Original data from Fisher et al. [4]. The statistical results of the different regressions are given in the electronic supplementary material, table S1. N total = 126 species where we had data for both number of cell types and total number of cells.

10 times more cell types than fungi [10], and it is unclear what can explain these differences. There are good reasons to speculate that the environment could be an important factor shaping the first trajectories of multicellular evolution with lasting consequences for later elaborations. First, the environment itself could determine the way in which multi-cellular groups form and hence relatedness between cells. Bonner [11] observed that clonal group formation, where daughter cells remain attached to mother cells after division, seems to be more common in lineages that originated in the sea compared to species that originated on land. If this is the case, it would mean that the environment where multicel-lularity originates could have profound consequences for subsequent evolutionary possibilities. Second, the physical constraints associated with living in water or on land are likely to affect many aspects of phenotypic evolution, for example, the need for support and structural reinforcement tissues, the diversity of dispersal mechanisms, and the biomechanics needed to sustain active motility.

Scaling theories provide powerful tools to test for such constraints, as an organism's body size can accurately predict complex traits such as metabolic rate, lifespan and growth rate [12], and because the shapes of these relationships can reflect fundamental physiological constraints on how diverse organisms can evolve [13]. Scaling relationships can also reveal outlier taxa where evolutionary innovation fuelled the breaking of ecological and physiological constraints [14]. In practice, scaling parameters (i.e. the slope ($b$) and intercept ($a$) in the equation $y = aM^b$) represent mechanistic hypotheses that, for the purpose of this study, relate the number of cell types ($y$) to the total number of cells ($M$). Isometric scaling ($b = 1$) provides a (probably unattainable) theoretical upper limit on cell diversification rates, as cell type and cell number would increase at the same rate (i.e. every added cell is a new type). Below this upper limit, allometric scaling ($b < 1$) would indicate that cell type increases at a slower rate than cell number (i.e. small organisms have more cell types relative to their body size).

There is a need to build on our understanding of the funda-mental factors influencing multicellular evolution—primarily the role of relatedness—and extend this to a broader perspective incorporating the role of the environment. The objectives of this paper are to (i) describe the variation in multicellular organiz-ational complexity across 139 species by investigating the scaling relationships between body size (total number of cells) and number of cell types; (ii) use phylogenetically controlled comparative analyses across 14 independent multicellular tran-sitions to assess the extent to which the environment determines how multicellular groups form and the consequences for whether obligate multicellularity could evolve; and (iii) test whether constraints imposed by the environment can explain why some lineages have reached higher levels of organizational complexity than others and can account for part of the variation in cell type diversity and differences in scaling relationships.

## 2. Results

### (a) Describing variation in body size and complexity

Across the species in our dataset, representing 14 independent transitions to multicellularity, we found that the scaling of cell type and cell number is strongly allometric (ordinary least-squares (OLS) regression): slope = 0.11, confidence inter-val = 0.10–0.13, $R^2 = 0.64$, figure 1a,b). This means that despite a positive association, the number of cell types increases much more slowly with cell number than arithmetic propor-tionality (i.e. isometric scaling) would predict. In other words, small organisms are organizationally more complex for their size than large organisms.

We next found evidence of a transition along the continuum of body size where the scaling relationship between number of cell types and total number of cells changed (electronic sup-plementary material, tables S1 and S2). We used a statistical approach to estimate this breakpoint in the regression at $6.3 \times 10^4$ total number of cells, corresponding to approximately six cell types. Smaller species (before the breakpoint) showed an

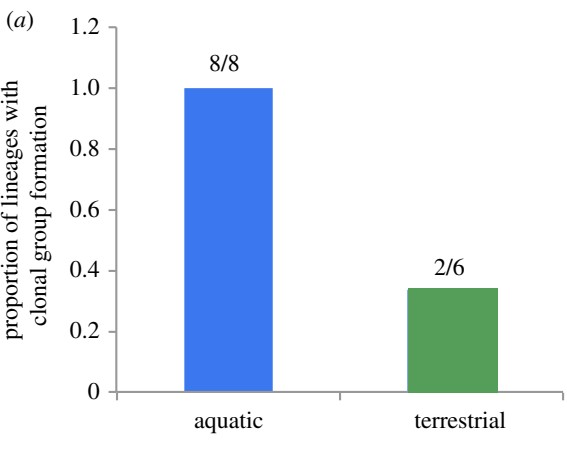

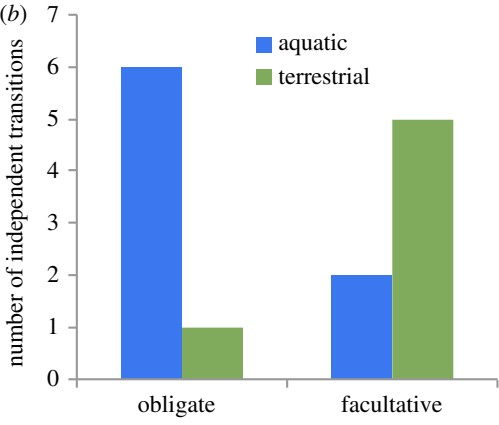

**Figure 2.** The origins of multicellularity in different environments. (*a*) The proportion of lineages that have clonal group formation that originated in aquatic and terrestrial environments. All multicellular lineages that originated in the sea have clonal group formation (8 out of 8 lineages), whereas only some of the multicellular lineages that originated on land have clonal group formation (2 out of 6). (*b*) Multicellular lineages that originated in water more commonly evolve obligate multicellularity (6 out of 8 lineages) compared to lineages that originated on land, which more often remain facultatively multicellular (5 out of 6 lineages).

allometric slope about three times as steep (OLS: slope = 0.21, confidence interval = 0.16 – 0.26, $R^2 = 0.61$) as larger species (above the breakpoint) (OLS: slope = 0.07, confidence interval = 0.03 – 0.11, $R^2 = 0.15$). This opens up the possibility that organisms face additional constraints in the accumulation of new cell types as they evolve larger body sizes and supports the observation that lineages consisting of small species gain new cell types more quickly as they evolve larger body sizes.

We further sought to understand the substantial reduction in cell type variation explained by cell number for large organisms (i.e. $R^2 = 0.15$) using a technique called quantile regression. Regression through the upper 90% quantile of the dataset suggests that there is an upper threshold to the number of cell types a species can have for its size, whereas there is substantial variation in the number of cell types below that threshold (figure 1*a*, dashed lines). This suggests that there could be other factors limiting the number of cell types below that threshold and these other limiting factors are especially important in larger species as the slope describing this upper limit is about half as steep ($b = 0.13$) than the upper limit for small species ($b = 0.25$) (electronic supplementary material, table S2).

### (b) The origins of multicellularity in different environments

Our results show that the physical environment (whether or not ancestral lineages lived in the water or on land) has had a major impact on both the origins and subsequent elaborations of multicellularity, both in determining how multicellular groups originally formed and how organizational complexity subsequently evolved.

We found that lineages in aquatic environments were significantly more likely to form multicellular groups through daughter cells remaining attached to mother cells after division (clonal group formation) (MCMCglmm, difference between aquatic and terrestrial: posterior mode = 5.74, credible intervals (CI) = 2.91 – 9.79, $p_{diff} = 0.0008$, $N_{species} = 139$; figure 2*a*). All of the multicellular lineages in our dataset that have their origins in water form multicellular groups in this way, whereas two thirds of the lineages that originated on land form groups through aggregation (non-clonal group formation) (figure 2*a*). In fact, there are only two lineages where multicellularity originated on land that employ clonal group formation—the Fungi and the plasmodial slime moulds and these tend to

grow in terrestrial environments of saturated humidity. This result confirms Bonner's original observation that clonal group formation is more common in multicellular lineages originating in the sea [11].

Second, we found that the transition to obligate multicellularity was significantly more likely to occur in aquatic environments compared to on land. Most (5 out of 6) lineages that evolved multicellularity on land remained facultatively multicellular (difference between aquatic and terrestrial: posterior mode = 6.59, CI = 4.29 – 8.72, $p_{diff} = < 0.0001$, $N_{species} = 139$; figure 2*b*). The only multicellular lineage that has evolved obligate multicellularity on land is the Fungi. This is consistent with this lineage also being a rare example of clonal group formation that originated on land, as the resulting clonal relatedness between cells is significantly associated with the transition to obligate multicellularity [4].

### (c) Multicellular organizational complexity on land versus in water

We found that the number of cell types of multicellular species currently found on land was significantly higher than those currently found in aquatic environments (figure 3), while controlling for the total number of cells (posterior mode = −0.77, CI = −1.42 to −0.11, $p_{diff} = 0.02$, $N_{species} = 121$; figure 4). The average number of cell types for aquatic lineages is eight whereas for terrestrial lineages it is 25. Species on land were, however, not significantly larger in size than those found in the sea (posterior mode = −2.79, CI = −9.04 to 1.81, $p_{diff} = 0.12$, $N_{species} = 121$). Overall, there was a significant phylogenetic correlation between number of cell types and total number of cells, meaning that species with more cell types also tend to be bigger owing to their shared ancestry (posterior mode = 0.90, CI = 0.72 to 0.96, $p_{diff} = < 0.0001$, $N_{species} = 121$). However, we also found a significant phenotypic correlation between these two variables, meaning that the association is also a result of a shared environment (posterior mode = 0.56, CI = 0.19 to 0.76, $p_{diff} = 0.004$, $N_{species} = 121$).

## 3. Discussion

We were interested in how multicellular complexity scales with body size and the role the physical environment could play in

*Proc. R. Soc. B* **287**: 20192963

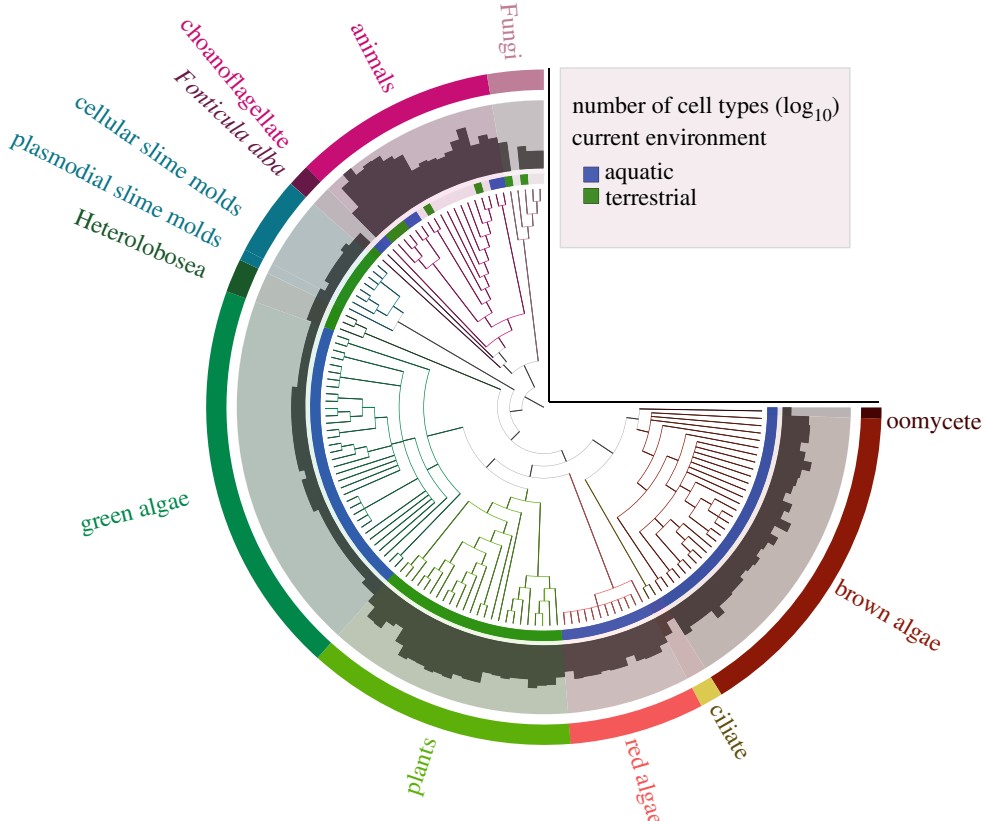

**Figure 3.** Phylogeny of the multicellular lineages in our dataset. Each lineage that independently evolved multicellularity within the eukaryotes is highlighted in a different colour—with two independent lineages within the ciliates ($N = 14$). The number of cell types ($\log_{10}$) from 1 – more than $10^{2.3}$ cell types are shown on the bar plots around the circumference of the phylogeny, and the current environment of each species (blue = aquatic, green = terrestrial) are also shown. The phylogeny was created using ANVI (anvi-server.org).

shaping the course of multicellular evolution. Overall, we found that the number of cell types scales allometrically with the total number of cells (echoing Bonner's observations [1]), and that the specific scaling relationship appears to change as organisms evolved larger body sizes. Our comparative analyses also show that the environment (aquatic or terrestrial) has a crucial impact on the trajectory of multicellular evolution. First, we found that clonal group formation giving rise to obligate multicellularity is significantly more common in lineages that evolved in aquatic environments. Second, we showed that current environmental conditions have an impact on multicellular evolution, with species living on land having a higher number of cell types compared to species found in aquatic environments.

Bonner [11] observed that clonal group formation was more common in multicellular lineages that evolved in the sea whereas aggregation was more common in terrestrial lineages. Our results provide formal support for this observation by including additional lineages and using phylogenetically controlled comparative analyses. Bonner speculated that this pattern could be because of water currents, meaning that cells in water need to stick together after they divide if they want to reap the benefits of being in a group [11]. This is not the case on land, where cells must use active motility (e.g. cilia, flagella, amoeboid movement) in order to form multicellular groups. It is clear from the species in our dataset that the multicellular lineages found on land (cellular slime moulds, ciliates, Fonticulidia, acrasid slime moulds) all have some form of motile cell stage, with a possible exception being the unicellular Fungi that have an ancestral aquatic lineage with motile cells [15]. It seems plausible that the biophysics of

moving through air and water has had a profound impact on the way in which multicellular groups could form on land and in the sea.

The data included in this study have some potential drawbacks that warrant a critical perspective. First, it is likely that our dataset underestimates the number of lineages that have facultative multicellularity. These species have transient multicellular phenotypes and many such species are still identified as unicellular. For example, *Saccharomyces cerevisiae* has a variety of multicellular phenotypes [16] and yet is still often not recognized as being facultatively multicellular [17]. Other lineages, notably the green algae, have evolved facultative multicellularity multiple times [18] and there are probably new examples to be found on land as well. However, it is unlikely that we have underestimated the number of lineages with obligate multicellularity. This is because there is some evidence that these species tend to be bigger and therefore more visible and complex [4] and potentially better studied [19,20]. Furthermore, there is no obvious reason to assume that under- or overestimation would be biased towards terrestrial or aquatic species. By inflating the number of facultative lineages, we would therefore not alter the pattern and the result we find—that obligate multicellularity has evolved much more often in water compared to on land. Second, the dataset underlying this study comes from a limited set of studies (notably [10]), which each understandably have errors in measurement of the number of cell types, total number of cells and in how a 'cell type' is defined. Defining cell types based on gene expression data would be ideal and potentially lead to many more cell types being described, but it is likely this increase in number of cell

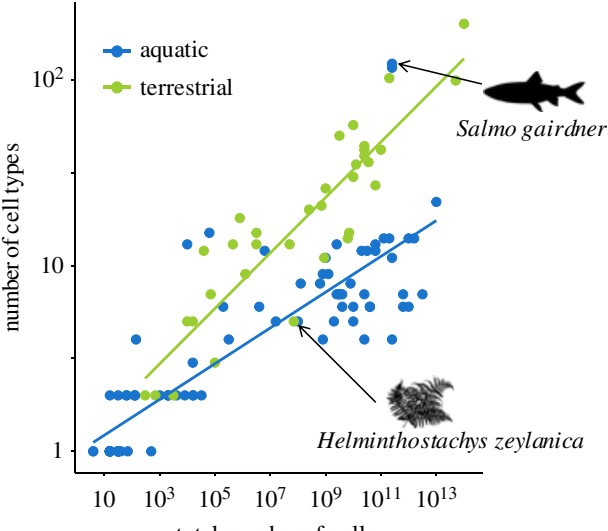

**Figure 4.** Organizational complexity and environmental constraints. Multicellular complexity, measured as both the number of cell types and the total number of cells, for species currently found in aquatic and terrestrial environments. Two notable outliers (*Salmo gairdneri* and *Helminthostachys zeylanica*) are highlighted with black arrows and images. *Salmo gairdneri* (rainbow trout) has an estimated 116 cell types and a total size of $2.51 \times 10^{11}$ cells, which is more similar to the terrestrial *Mus musculus* (mouse) than to other aquatic species. *Helminthostachys zeylandica* (a member of the fern family) has much lower complexity for its size than other terrestrial multicellular species ( just five cell types). These data have been taken from the dataset of [10].

types would be consistent across taxa, therefore minimally affect the fundamental patterns we see. Broad comparative studies, such as this one, almost always rely on using a large assembly of data of variable accuracy.

Not only does the environment affect how multicellular groups form, but we show that it also has a major impact on the scaling relationships between size and complexity. Species that live on land tend to be more complex for their size compared to species that live in water (i.e. with a higher slope, figure 4) and this could be for several reasons. Land dwelling organisms need more support structures than their aquatic counterparts—this is because water provides natural support through buoyancy whereas air does not. Organisms living on land therefore needed to increasingly invest in stems and skeletons to 'hold themselves' up as their body size increased (i.e. skeleton mass $\sim M^{b > 1}$, [12]), possibly leading to greater diversification of cell types and tissues than required by organisms in the sea. This scaling logic can further be extended to resource allocation dynamics within organisms (e.g. vascular networks), although systematic effects on cell diversity and differences between land and water remain to be elucidated. There are also other potentially confounding physiological parameters, for example, the possibility that autotrophic lineages compete for light, both on land and in the water, whereas heterotrophic lineages do not. Assuming such competition has selected for support tissues with specialized cell types, these are more costly to maintain on land (e.g. rainforest trees versus kelp forest).

Our study reveals intriguing outliers that deserve further consideration. The Fungi are a strikingly unusual (mostly) multicellular lineage that present particular challenges in this study for several reasons. Our dataset includes only three fungal species, that do not accurately reflect the true level of fungal complexity in nature (e.g. *Neurospora crassa*, which has 28 distinct cell types [21]). The Fungi are now also known to have independently evolved complex multicellularity up to 11 times [22]. However, we lacked complete data on the number of cells and cell types to allow inclusion of more lineages in the analyses of complexity presented here (figure 4). While the lack of data is a problem for drawing detailed conclusions about scaling relationships within fungi, our phylogenetic analyses would be unaffected (as they deal with independent contrasts between lineages, not species *per se*) so we can be confident in our conclusion that clonality is associated with aquatic ancestral environments (figure 2*a*). Second, it has been argued [22,23] that multicellular group formation in the Fungi cannot be strictly classified as clonal or aggregative, owing to the way in which multi-nucleate fungal hyphae form, and this is a potential limitation of our study where we had to classify every species in these two discrete categories. Finally, it is also difficult to confidently class fungi as either terrestrial or aquatic, as they live and have evolved mostly at the air–water interface. Perhaps a closer look at the Fungi as putative 'exceptions to the rule' could help to unravel the generality of the relationship between the environment and multicellular complexity that we uncovered.

Our results, showing that the physical environment has impacted the way in which multicellular groups form, could therefore shed light on the role of the environment for other major evolutionary transitions [24,25] and help us understand the balance between how intrinsic and extrinsic factors can affect evolutionary trajectories.

# 4. Material and methods

## (a) Data collection

The data used in this study were originally published in Fisher *et al.* [4] and are stored in the Dryad Digital Repository (original data can be found here: https://datadryad.org/resource/doi:10.5061/dryad.27q59). The full data, code and phylogeny used in this study can also be found on Dryad (https://doi.org/10.5061/dryad.dv41ns1vn [26]). In summary, we conducted an extensive literature search on multicellular species, searching specifically for information on multicellular complexity (number of cell types and the total number of cells), the ways in which groups formed and whether or not they were obligately or facultatively multicellular. We focus only on the eukaryotic species included in Fisher *et al.* [4], as it is unfeasible to apply the same standards of allocating either 'aquatic' or 'terrestrial' to prokaryotes. Our full dataset can be found in the electronic supplementary material, table S6. Information of both the number of cell types and total number of cells (allowing us to estimate 'complexity') was essential for analyses where we included complexity (electronic supplementary material, table S5) and we therefore focused our data collection effort on species where information on both these traits was available.

The method of defining a 'cell type' is an ongoing challenge and one which we do not claim to have solved in this paper. The cell type data we use are mainly based on morphological characteristics, where a definition based on detailed gene expression data would be more desirable. In their original paper, Bell & Mooers [10] highlight the problem with defining a cell type—'In short, (the definition of cell types) is deplorably inexact, but we defend them as being essential in a preliminary treatment of an important problem that has not hitherto been approached quantitatively at all' [10, p. 347].

**Table 1.** At least 14 transitions to multicellularity occurred within the eukaryotes. (Estimates of the number of independent transitions to multicellularity in each lineage are given along with the environment where the lineage originated when it evolved multicellularity, average number of cell types and the corresponding references. We have not included two other known transitions to multicellularity—the diatoms [28] and *Sorodiplophrys* (Stramenopiles) [29]—due to a lack of data on cell types and environment of origin. See figure 3.)

| lineage | estimated number of transitions | average number of cell types | ancestral environment where multicellularity evolved | obligate or facultative multicellularity | reference(s) |
|---|---|---|---|---|---|
| acrasid slime moulds | 1 | 2 | terrestrial | facultative | [30] |
| brown algae | 1 | 6.9 | aquatic | obligate | [19,31] |
| cellular slime moulds | 1 | 2 | terrestrial | facultative | [32] |
| chlorophyte algae | 1–4 | 1.5 | aquatic | facultative & obligate | [18] |
| choanoflagellates | 1 | 1 | aquatic | facultative | [33] |
| ciliates | 2 | 2.5 | terrestrial & aquatic | facultative & obligate | [34–36] |
| *Fonticula alba* (Fonticulida) | 1 | 2 | terrestrial | facultative | [37] |
| Fungi | 1+[a] | 7 | terrestrial | obligate | [27,38] |
| Metazoa | 1 | 101.6 | aquatic | obligate | [27] |
| oomycetes | 1 | 1 | aquatic | facultative & obligate | [39] |
| plants | 1 | 22.2 | aquatic | obligate | [27] |
| plasmodial slime moulds | 1 | 2 | terrestrial | facultative | [40] |
| red algae | 1+ | 10.8 | aquatic | obligate | [27] |

[a]The Fungi evolved obligate multicellularity on land twice—once in the Ascomycota and once in the Basidiomycota [19,27,41]. However, we only have examples of Ascomycota in our dataset, so write this as '1+' here, to avoid inconsistency with the number of transitions in figure 2.

In this study, we expanded on the eukaryotic species in the original dataset by adding information on the ancestral and current environment of each species. We considered any species found on land as terrestrial and any species found in freshwater, brackish or marine environments as aquatic. We found information about the current environment of a species by searching on Google Scholar for publications and also taxa-specific websites, such as AlgaeBase and WoRMS. Where there was only information about ancestral or current environment at a higher taxonomic level (i.e. at the family level but no generic or species information), we assumed it was the same environment for the species in our dataset. We found information on the ancestral environment of each species through broad reviews on the origins of multicellularity including [11,18,27]. It is important to stress that we were interested in the ancestral environment *when multicellularity evolved* and therefore that was not always the same as the ancestral environment for the whole lineage, including unicellular groups (e.g. for the Fungi, [15]).

Of the 139 species in the dataset, 18 species had a terrestrial ancestral environment and 121 species had an aquatic ancestral environment. For the current environment, 84 species are aquatic, 43 are terrestrial and 12 are unknown. Our full dataset included 139 species but only 121 of these species had complete data on the number of cell types, total number of cells, current environment, ancestral environment and the mode of group formation.

## (b) Independent transitions to multicellularity
Using information from published papers, we identified that within the eukaryotes there have been at least 14 independent transitions to multicellularity (both facultative and obligate) (table 1 and figure 3). However, we have most likely underestimated the number of transitions in several groups owing to uncertainty about the number of independent transitions within them. For example, it is thought that there have been at least two transitions to obligate multicellularity within the Fungi [27,38] and multiple transitions to facultative multicellularity in the green algae [18] and in the red algae [19]. Therefore, our analyses are conservative and assumed just one transition within each group.

## (c) Statistical methods
### (i) Scaling relationships
As a first step in analysing the data, we performed a series of OLS regressions to estimate $a$ and $b$ in the scaling equation $\log_{10}y = \log_{10}a + b\log_{10}M$ meant to describe the dependence of the number of cell types, $y$ on the total number of cells ($M$). From these regressions, we evaluated the existence of allometry and estimate the intercept and slope in the scaling of $\log_{10}$(cell type) against $\log_{10}$(cell number) across all data. We used OLS regression rather than reduced major axis regression (RMA) even though our $X$ variables contained measurement error, based on the $X$–$Y$ symmetry principle of Smith (2009). We note however, that OLS and RMA approaches yielded very similar results, just with slightly steeper slopes in all cases. We then used the package 'segmented' in R [42] to test if there is a 'breakpoint' in the regression—the point at which the shape of the relationship changes abruptly. We then used the OLS approach described above to evaluate scaling relationships on either side of the breakpoint.

We also noted that the scatter plots producing average scaling relationships appeared triangular and thus hypothesized that they reflected a constraint function such that total number of cells is necessary, but not sufficient to explain variation in number of

cell types [43,44]. To test this hypothesis, we used least quantile regressions to describe scaling for the upper ninetieth quantiles of the overall plot and separately for scaling relationships on either side of the breakpoint [45,46].

### (ii) Bayesian analyses

We used the statistical package MCMCglmm [47] to run Bayesian general linear models with Markov chain Monte Carlo (MCMC) estimation. We fitted three models. First, we tested whether the environment affected the way in which multicellular groups form by fitting a model with group formation as a categorical response variable and the ancestral environment as a categorical explanatory variable (electronic supplementary material, table S3). Second, we tested whether the environment affected the likelihood of obligate or facultative multicellularity by fitting a model with obligate/facultative as a categorical response variable and the ancestral environment as a categorical explanatory variable (electronic supplementary material, table S4).

Finally, we tested whether multicellular complexity differed between lineages in terrestrial versus aquatic environments by fitting a multi-response model with several explanatory variables using the number of cell types and the logarithm of total number of cells as Poisson and Gaussian response variables, respectively (electronic supplementary material, table S4). This allowed us to use both number of cell types and the total number of cells as a combined measure of multicellular complexity, rather than having to run several analyses using different response variables. We fitted several categorical fixed effects: the current environment (aquatic or terrestrial), whether the species is obligately or facultatively multicellular, and the mode of group formation (non-clonal or clonal) to control for the known effects of group formation and obligateness on multicellular complexity [4].

In the first two models, we used uninformative inverse-gamma priors because we had a categorical response variable. We also fixed the residual variance to 1 and specified family = categorical. In the final model, we used uninformative priors because we had a multi-response model with both Poisson and Gaussian response variables and categorical explanatory variables. We ran the models for 6 000 000 iterations, with a burn-in of 1 000 000 and a thinning interval of 1000. These were the values that optimized the chain length while also allowing our models to converge, which we assessed visually using variance-covariance matrix traceplots. We then ran each model three times and used the Gelman–Rubin diagnostic to quantitatively check for convergence. We assumed that our models had converged when the phylogenetic signal-representation was less than 1.1.

We calculated the correlations between the number of cell types and the total number of cells, i.e. cov(number of cell types, total number of cells)/sqrt(var(number of cell types) × var(total number of cells) for species in different environments. We tested if the correlation was significantly different between environments by examining if the 95% CI of the difference between the correlations spanned 0 and calculating the per cent iterations where the correlation for species living in aquatic environments was greater than the correlation for those living on land.

### (iii) Phylogeny construction

We built the phylogeny for this study using the Open Tree of Life (opentreeoflife.org) [48–50], which creates synthetic trees built from published phylogenies and taxonomic information. We then used the R package 'rotl' that interacts with the online database and constructs phylogenies (https://cran.r-project.org/web/packages/rotl/index.html) [51]. For the majority of species in our dataset, the exact species was also present in a published phylogeny so we could use phylogenetic information about that species. However, for a few species that were not present in the Open Tree of Life dataset, we had to assign instead a closely related species in the same genus or use a family-level classification. Owing to the fact that most species in our dataset represent phylogenetically distant groups on the eukaryotic tree and our phylogeny does not include branch lengths, we are confident that this compromise did not affect our statistical analysis. The phylogeny presented in figure 3 was created for visual purposes using ANVI (anvi-server.org).

Data accessibility. Data are available in the electronic supplementary material and from the Dryad Digital Repository: https://doi.org/10.5061/dryad.27q59 [26].

Authors' contributions. R.M.F. collected the data; R.M.F. and J.Z.S. analysed the data; R.M.F., J.Z.S. and J.J.B. wrote the manuscript.

Competing interests. We declare we have no competing interests.

Funding. R.M.F. was supported by a Carlsberg Distinguished Post-doctoral Fellowship (CF16-0336) hosted by J.J.B. and J.Z.S. was supported by a European Research Council Starting grant (no. ELEVATE 757810).

Acknowledgements. We thank Stuart West and Guy Cooper for thought-provoking discussions and comments and Stefania Kapsetaki, Jordan Okie, and Jamie Gillooly for helpful edits on a final version of the manuscript. We also thank three reviewers for their helpful comments that significantly improved this manuscript. We dedicate this paper to the memory of John Tyler Bonner.

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
