## [Reviewer comments · Proceedings of the Royal Society B: Biological Sciences]

Review History

RSPB-2019-2963.R0 (Original submission)

Review form: Reviewer 1

Recommendation

Major revision is needed (please make suggestions in comments)

Scientific importance: Is the manuscript an original and important contribution to its field?

Excellent

General interest: Is the paper of sufficient general interest?

Good

Quality of the paper: Is the overall quality of the paper suitable?

Good

Is the length of the paper justified?

Yes

Should the paper be seen by a specialist statistical reviewer?

Yes

Do you have any concerns about statistical analyses in this paper? If so, please specify them explicitly in your report.

Yes

It is a condition of publication that authors make their supporting data, code and materials available - either as supplementary material or hosted in an external repository. Please rate, if applicable, the supporting data on the following criteria.

Is it accessible?

Yes

Is it clear?

Yes

Is it adequate?

Yes

Do you have any ethical concerns with this paper?

No

Comments to the Author

The manuscript probes a previously published data set for a series of questions regarding multicellularity evolution. This data set spans a phylogenetically diverse set of eukaryotes, representative of the majority of known eukaryotic multicellular lineages.

Firstly, the authors use scaling theory to quantify the association between the organism size (measured by the mean number of cells of an organism) and its complexity. As a proxy for complexity, they rely on the number of cell types in each species. They conclude that the relationship between the number of cell types and the total number of cells in an organism follows two different regimes: one for small and another for large organisms.

The second question analyzed is the effect of the environment of origin in the process of group formation (clonal or non-clonal). Their phylogenetically informed approach corroborates Bonner's observation that clonal group formation is more common when the transition happens in an aquatic environment.

Finally, the authors look into the complexity level of multicellular organisms on land vs in water. They find that the complexity is in general higher for land organisms, a result that remains strong even after controlling for phylogenetic factors.

The manuscript is clear and engaging to read, and the issue investigated is relevant. I particularly like the strong statistical confirmation that clonal group formation is more frequent for aquatic transitions. Nevertheless, I have some concerns regarding the scaling relations obtained.

Major comments

The authors conclude that two different power-law regimes exist, corresponding to small and large organisms. However, in my view, the evidence presented does not support this conclusion. The R^2 for the fitting to the full data is larger than the ones obtained for partial data in each subset, suggesting that a fit to the full data retains a relatively high explanatory power without requiring additional hypotheses.

Furthermore, the authors state that they used an R package to test for the existence of the breakpoint, but provide no test result.

Minor comments

- line 113-114: the value of R^2 from OLS is reported amidst the values of RMA fitting, which may be misleading

- line 375: is table S1 the correct reference?

- table S2: N is 126 for the small species. Is it not 50?

- table S2: no R^2 provided
- tables S3-5: no N provided
- the authors use reduced major axis regression with the argument that both X and Y variables have associated errors. It is not clear to me that this is the best approach to take (please see, e.g., RJ Smith, Use and misuse of the reduced major axis for line-fitting, 2009)
- I would like to see a comment on why the authors have only considered eukaryotes in this study, given that the original data set (Fisher et al., 2013) included species from all three domains of life

Review form: Reviewer 2 (Lazlo Nagy)

Recommendation

Accept with minor revision (please list in comments)

If yes, please enter your name here as it should appear.

Laszlo G. Nagy

Scientific importance: Is the manuscript an original and important contribution to its field?

Excellent

General interest: Is the paper of sufficient general interest?

Good

Quality of the paper: Is the overall quality of the paper suitable?

Excellent

Is the length of the paper justified?

Yes

Should the paper be seen by a specialist statistical reviewer?

Yes

Do you have any concerns about statistical analyses in this paper? If so, please specify them explicitly in your report.

No

It is a condition of publication that authors make their supporting data, code and materials available - either as supplementary material or hosted in an external repository. Please rate, if applicable, the supporting data on the following criteria.

Is it accessible?

Yes

Is it clear?

Yes

Is it adequate?

No

Do you have any ethical concerns with this paper?

No

Comments to the Author

Fisher et al test an old hypothesis by Bonner that postulates that the environment determines the type and stability of multicellular associations that emerge. I found the paper very interesting,

concise and the analyses robust. I have minor suggestions to the authors that they may take or leave.

The aspect that made me think the most is how reliable the input data are that the authors took from Bell and Mooers 1997? Looking at the table in this publication, I have the feeling that its both taxonomically biased and inaccurate in many cases. I'm not questioning Bell-Mooers' efforts in compiling those data, but state of the art at the time might not have allowed them to reach high accuracy in many cases. I can only assess fungi, where I feel serious limitations. I'm not sure what the authors could do to update these figures or to mitigate their potential bias on their results, but at least some discussion would be useful.

As a general suggestion I think the data used in the analyses could be made more accessible by including it in the main text. Consider moving Figure S1 to the main text and displaying the examined traits next to taxon names (like in a circos plot).

Specific comments:

l34 - wild type yeast under normal condition does not form aggregates (it forms biofilms, but that's not an aggregation).

l55 - as a mycologist I argue that ref10 has outdated numbers for fungal cell type diversity, which can reach 30 (which is based only on simple morphological observations, Kues 2015 Fungal Biol Rev., Bistis & Read 2003 Fungal Genet Biol.)

l93-95 - the sentence reads like it has a bit of circularity in logic.

l234 - As an alternative explanation, I would also argue that water currents distribute nutrients to all the cells of a clonal colony (in a sponge for example), whereas goods are more patchy on land and therefore need to be actively foraged for.

l237 - please mention the containing lineages not example genera

l239 - yes, but fungi with motile cells are connected to water. I would argue that multicellular fungi don't have a motile cell type, because fast apical growth of hyphae enables foraging for nutrients.

l246 - Kiss et al (Nat Comms 2019) argues in a different way for facultative multicellularity in yeasts, the conservation of multicellularity-related genes and facultative hypha formation.

l250 - I don't see a direct link between being obligate and being big. Please rephrase.

l270 - there is some discussion on that in the below mentioned papers and in Nagy et al Microb Spectrum 2017. "Hyphal multicellularity" indeed seems to be an exception from many rules, and it's not clearly clonal in the classic sense (clonal at the level of nuclei, not at cells).

l279 - possibly?

l301 - finishing the discussion with a broad conclusion on multicellularity would probably read better to me than a somewhat forced generalization to completely unrelated organismal traits.

l322 - where was data for the ancestral environment of lineages taken? Ideally, this attribute could be inferred by ancestral state reconstructions.

l341 - see Kiss et al Nat Comms 2019 and Nagy 2018 Biol Reviews on an update on fungi. The transition to simple multicellularity likely occurred once, whereas complex multicellularity emerged several times.

Laszlo G. Nagy

Review form: Reviewer 3

Recommendation

Accept with minor revision (please list in comments)

Scientific importance: Is the manuscript an original and important contribution to its field?

Good

General interest: Is the paper of sufficient general interest?

Good

Quality of the paper: Is the overall quality of the paper suitable?

Good

Is the length of the paper justified?

Yes

Should the paper be seen by a specialist statistical reviewer?

No

Do you have any concerns about statistical analyses in this paper? If so, please specify them explicitly in your report.

Yes

It is a condition of publication that authors make their supporting data, code and materials available - either as supplementary material or hosted in an external repository. Please rate, if applicable, the supporting data on the following criteria.

Is it accessible?

Yes

Is it clear?

Yes

Is it adequate?

Yes

Do you have any ethical concerns with this paper?

No

Comments to the Author

Report on "The evolution of multicellular complexity: the role of relatedness and environmental constraints" by RM Fisher et al.

The paper has a compelling history, a fair amount of data analysis and metanalysis. The authors test the hypothesis (by John T. Bonner) that the evolution of multicellularity is affected by where such transition occur: in an aquatic vs. terrestrial environment. In particular, what is interesting is that clonality vs. aggregativity differ significantly in the number of cell types (and organismal size), which could be well explained by how the environment favored one or the other type of transition to multicellularity. By doing an extensive meta-analysis of the literature, the authors concluded that the theory of social evolution (relatedness, in particular) cannot completely account for these differences, and that the environment has played a major role. I believe that the paper makes an important contribution to the understanding of the transition to multicellularity, and it also brings forth a novel perspective in scaling relationships that (so far) have been seldom studied in the literature on multicellular evolution.

I would suggest the following discussion/clarification points:

1. Why is the isometric scaling the frame of reference (the null hypothesis)? It seems unrealistic that any organism would ever evolve one cell type directly proportional to its cell count. Thus, a null model that cannot ever exist(?), even in principle, might not be that useful.
2. Is there a conflict in the way different authors in the papers cited conceive (and account for) a "cell type"? If so, how is this addressed in a way that makes the conclusions robust to such discrepancies?
3. Considering the wide range of organisms studied, it also seems a sensible discussion to mention how the developmental stage at which the cell types and cell number were "counted" could affect the conclusions. Are these cell types the total number of types present throughout the life cycle? Are they the types at a "fully developed" organism? How these temporal/developmental boundaries can be compared across such diverse groups of organisms?
4. I understand that one can break a regression into multiple sections, but why was 2 groupings the chosen one? (at 10^4 cells). Even though it gives a 'nice' discussion between "small" vs. "big" organisms, it seems a bit arbitrary. Was it sensitive to the sampling distribution (the small

organisms seem to be “overrepresented” at ~4 cell types)? At the boundary, are there organisms qualitatively distinct? (i.e. close taxa grouped in one or the other groups).

5. Even though there is a discussion of sampling bias at the end, could the authors test the robustness of the main conclusions to sampling biases? (i.e. some bootstrap method on the datasets, weighted perhaps by taxa?).

6. Some of the data, especially on the number of independent transitions to multicellularity, is not current with the latest phylogenetic developments. Specifically, the 'red algae' are described as having 1+ transitions to multicellularity in Figure 1 (citing Andy Knoll's 2011 review paper)- these two papers say either 2 or 3 times:

Cock JM, Collén J (2015) Independent emergence of complex multicellularity in the brown and red algae in *Evolutionary Transitions to Multicellular Life*. (Springer), pp. 335–361.

Yoon HS, Müller KM, Sheath RG, Ott FD, Bhattacharya D (2006) Defining the major lineages of red algae (rhodophyta). *Journal of phycology* 42(2):482–492.

Perhaps more importantly, the fungi have recently come into focus much more than before. The old estimate was 2 origins, but Lazlo Nagy's groundbreaking 2018 paper showed it was likely between 8 and 11.

Nagy LG, Kovács GM, Krizsán K (2018) Complex multicellularity in fungi: evolutionary convergence, single origin, or both? *Biological Reviews* 93(4):1778–1794

Small observations:

1. In Fig. 1, the X-axis legend in the left panel reads “Total number of cells” and the right panel reads “Total number of cells (log)”. I'd keep the legends consistent.
2. X-axis legend of Fig. 3 (right panel) seems to have formatting errors (with 10^{11} and 10^{13}).

Decision letter (RSPB-2019-2963.R0)

03-Feb-2020

Dear Dr Fisher:

Your manuscript has now been peer reviewed and the reviews have been assessed by an Associate Editor. As you will see, the reviewers and the AE have raised some concerns with your manuscript and we would like to invite you to revise your manuscript to address them. These are outlined nicely below, although I would add to the AE's comments that I would like you to address reviewer 2 and 3's comments about the degree to which we can rely on the older data sets (i.e., Bell and Mooers 1997). Note that I am not recommending that you do not use these data, but comment on the degree to which these were limited by the techniques available at the time they were published. Of course, please also address each of the reviewer's other comments. These comments (not including confidential comments to the Editor) and the comments from the Associate Editor are included at the end of this email for your reference.

When submitting your revision please upload a file under "Response to Referees" - in the "File Upload" section. This should document, point by point, how you have responded to the reviewers' and Editors' comments, and the adjustments you have made to the manuscript. We

require a copy of the manuscript with revisions made since the previous version marked as 'tracked changes' to be included in the 'response to referees' document.

Research ethics:

Use of animals and field studies:

Online supplementary material will also carry the title and description provided during submission, so please ensure these are accurate and informative. Note that the Royal Society will not edit or typeset supplementary material and it will be hosted as provided. Please ensure that

the supplementary material includes the paper details (authors, title, journal name, article DOI). Your article DOI will be 10.1098/rspb.[paper ID in form xxxx.xxxx e.g. 10.1098/rspb.2016.0049].

Please submit a copy of your revised paper within three weeks. If we do not hear from you within this time your manuscript will be rejected. If you are unable to meet this deadline please let us know as soon as possible, as we may be able to grant a short extension.

Best wishes,

Dr Sarah Brosnan
Editor, Proceedings B
mailto:proceedingsb@royalsociety.org

Associate Editor
Comments to Author:

All three peer reviewers agree that the paper makes an important contribution to the understanding of the transition to multicellularity, particularly in its incorporation of scaling relationships.

However, this manuscript requires revision to be appropriate for publication in Proc B. Specifically, multiple reviewers have concerns about the selection of the breakpoint between "large" and "small" organisms, and the statistical support for a model that estimates different power law regimes for each of these categories, as opposed to a single unified model. Please address this concern as well as the other comments and suggestions made by the individual reviewers. Thank you,

Reviewer(s)' Comments to Author:

Referee: 1

Comments to the Author(s)

The manuscript probes a previously published data set for a series of questions regarding multicellularity evolution. This data set spans a phylogenetically diverse set of eukaryotes, representative of the majority of known eukaryotic multicellular lineages.

Firstly, the authors use scaling theory to quantify the association between the organism size (measured by the mean number of cells of an organism) and its complexity. As a proxy for complexity, they rely on the number of cell types in each species. They conclude that the relationship between the number of cell types and the total number of cells in an organism follows two different regimes: one for small and another for large organisms.

The second question analyzed is the effect of the environment of origin in the process of group formation (clonal or non-clonal). Their phylogenetically informed approach corroborates Bonner's observation that clonal group formation is more common when the transition happens in an aquatic environment.

Finally, the authors look into the complexity level of multicellular organisms on land vs in water. They find that the complexity is in general higher for land organisms, a result that remains strong even after controlling for phylogenetic factors.

The manuscript is clear and engaging to read, and the issue investigated is relevant. I particularly

like the strong statistical confirmation that clonal group formation is more frequent for aquatic transitions. Nevertheless, I have some concerns regarding the scaling relations obtained.

Major comments

The authors conclude that two different power-law regimes exist, corresponding to small and large organisms. However, in my view, the evidence presented does not support this conclusion. The R^2 for the fitting to the full data is larger than the ones obtained for partial data in each subset, suggesting that a fit to the full data retains a relatively high explanatory power without requiring additional hypotheses.

Furthermore, the authors state that they used an R package to test for the existence of the breakpoint, but provide no test result.

Minor comments

- line 113-114: the value of R^2 from OLS is reported amidst the values of RMA fitting, which may be misleading
- line 375: is table S1 the correct reference?
- table S2: N is 126 for the small species. Is it not 50?
- table S2: no R^2 provided
- tables S3-5: no N provided
- the authors use reduced major axis regression with the argument that both X and Y variables have associated errors. It is not clear to me that this is the best approach to take (please see, e.g., RJ Smith, Use and misuse of the reduced major axis for line-fitting, 2009)
- I would like to see a comment on why the authors have only considered eukaryotes in this study, given that the original data set (Fisher et al., 2013) included species from all three domains of life

Referee: 2

Comments to the Author(s)

Fisher et al test an old hypothesis by Bonner that postulates that the environment determines the type and stability of multicellular associations that emerge. I found the paper very interesting, concise and the analyses robust. I have minor suggestions to the authors that they may take or leave.

The aspect that made me think the most is how reliable the input data are that the authors took from Bell and Mooers 1997? Looking at the table in this publication, I have the feeling that its both taxonomically biased and inaccurate in many cases. I'm not questioning Bell-Mooers's efforts in compiling those data, but state of the art at the time might not have allowed them to reach high accuracy in many cases. I can only assess fungi, where I feel serious limitations. I'm not sure what the authos could do to update these figures or to mitigate their potential bias on their results, but at least some discussion would be useful.

As a general suggestion I think the data used in the analyses could be made more accessible by including it in the main text. Consider moving Figure S1 to the main text and displaying the examined traits next to taxon names (like in a circos plot).

Specific comments:

134 - wild type yeast under normal condition does not form aggregates (it forms biofilms, but that's not an aggregation).

155 - as a mycologist I argue that ref10 has outdated numbers for fungal cell type diversity, which can reach 30 (which is based only on simple morphological observations, Kues 2015 Fungal Biol Rev., Bistis & Read 2003 Fungal Genet Biol.)

193-95 - the sentence reads like it has a bit of circularity in logic.

- As an alternative explanation, I would also argue that water currents distribute nutrients to all the cells of a clonal colony (in a sponge for example), whereas goods are more patchy on land and therefore need to be actively foraged for.

- please mention the containing lineages not example genera

- yes, but fungi with motile cells are connected to water. I would argue that multicellular fungi don't have a motile cell type, because fast apical growth of hyphae enables foraging for nutrients.

- Kiss et al (Nat Comms 2019) argues in a different way for facultative multicellularity in yeasts, the conservation of multicellularity-related genes and facultative hypha formation.

- I don't see a direct link between being obligate and being big. Please rephrase.

- there is some discussion on that in the below mentioned papers and in Nagy et al Microb Spectrum 2017. "Hyphal multicellularity" indeed seems to be an exception from many rules, and it's not clearly clonal in the classic sense (clonal at the level of nuclei, not at cells).

- possibly?

1301 - finishing the discussion with a broad conclusion on multicellularity would probably read better to me than a somewhat forced generalization to completely unrelated organismal traits.

1322 - where was data for the ancestral environment of lineages taken? Ideally, this attribute could be inferred by ancestral state reconstructions.

1341 - see Kiss et al Nat Comms 2019 and Nagy 2018 Biol Reviews on an update on fungi. The transition to simple multicellularity likely occurred once, whereas complex multicellularity emerged several times.

Laszlo G. Nagy

Referee: 3

Comments to the Author(s)

Report on "The evolution of multicellular complexity: the role of relatedness and environmental constraints" by RM Fisher et al.

The paper has a compelling history, a fair amount of data analysis and meta-analysis. The authors test the hypothesis (by John T. Bonner) that the evolution of multicellularity is affected by where such transition occurs: in an aquatic vs. terrestrial environment. In particular, what is interesting is that clonality vs. aggregativity differ significantly in the number of cell types (and organismal size), which could be well explained by how the environment favored one or the other type of transition to multicellularity. By doing an extensive meta-analysis of the literature, the authors concluded that the theory of social evolution (relatedness, in particular) cannot completely account for these differences, and that the environment has played a major role. I believe that the paper makes an important contribution to the understanding of the transition to multicellularity, and it also brings forth a novel perspective in scaling relationships that (so far) have been seldom studied in the literature on multicellular evolution.

I would suggest the following discussion/clarification points:

1. Why is the isometric scaling the frame of reference (the null hypothesis)? It seems unrealistic that any organism would ever evolve one cell type directly proportional to its cell count. Thus, a null model that cannot ever exist(?), even in principle, might not be that useful.
2. Is there a conflict in the way different authors in the papers cited conceive (and account for) a

“cell type”? If so, how is this addressed in a way that makes the conclusions robust to such discrepancies?

3. Considering the wide range of organisms studied, it also seems a sensible discussion to mention how the developmental stage at which the cell types and cell number were “counted” could affect the conclusions. Are these cell types the total number of types present throughout the life cycle? Are they the types at a “fully developed” organism? How these temporal/developmental boundaries can be compared across such diverse groups of organisms?

4. I understand that one can break a regression into multiple sections, but why was 2 groupings the chosen one? (at 10^4 cells). Even though it gives a ‘nice’ discussion between “small” vs. “big” organisms, it seems a bit arbitrary. Was it sensitive to the sampling distribution (the small organisms seem to be “overrepresented” at ~ 4 cell types)? At the boundary, are there organisms qualitatively distinct? (i.e. close taxa grouped in one or the other groups).

5. Even though there is a discussion of sampling bias at the end, could the authors test the robustness of the main conclusions to sampling biases? (i.e. some bootstrap method on the datasets, weighted perhaps by taxa?).

6. Some of the data, especially on the number of independent transitions to multicellularity, is not current with the latest phylogenetic developments. Specifically, the 'red algae' are described as having 1+ transitions to multicellularity in Figure 1 (citing Andy Knoll's 2011 review paper)- these two papers say either 2 or 3 times:

Cock JM, Collén J (2015) Independent emergence of complex multicellularity in the brown and red algae in *Evolutionary Transitions to Multicellular Life*. (Springer), pp. 335–361.

Yoon HS, Müller KM, Sheath RG, Ott FD, Bhattacharya D (2006) Defining the major lineages of red algae (rhodophyta). *Journal of phycology* 42(2):482–492.

Perhaps more importantly, the fungi have recently come into focus much more than before. The old estimate was 2 origins, but Lazlo Nagy's groundbreaking 2018 paper showed it was likely between 8 and 11.

Nagy LG, Kovács GM, Krizsán K (2018) Complex multicellularity in fungi: evolutionary convergence, single origin, or both? *Biological Reviews* 93(4):1778–1794

Small observations:

1. In Fig. 1, the X-axis legend in the left panel reads “Total number of cells” and the right panel reads “Total number of cells (log)”. I’d keep the legends consistent.
2. X-axis legend of Fig. 3 (right panel) seems to have formatting errors (with 10^{11} and 10^{13}).

Author's Response to Decision Letter for (RSPB-2019-2963.R0)

See Appendix A.

Decision letter (RSPB-2019-2963.R1)

08-Jun-2020

Dear Dr Fisher

I am pleased to inform you that your manuscript RSPB-2019-2963.R1 entitled "The evolution of multicellular complexity: the role of relatedness and environmental constraints" has been accepted for publication in Proceedings B pending minor revisions. These are listed below, in the comments from the Associate Editor and the reviewers. I invite you to respond to the referee(s)' comments and revise your manuscript. Because the schedule for publication is very tight, it is a condition of publication that you submit the revised version of your manuscript within 7 days. If you do not think you will be able to meet this date please let us know.

Sincerely,

Dr Sarah Brosnan
Editor, Proceedings B
<mailto:proceedingsb@royalsociety.org>

Associate Editor:
Board Member: 1

Decision letter (RSPB-2019-2963.R2)

25-Jun-2020

Dear Dr Fisher

I am pleased to inform you that your manuscript entitled "The evolution of multicellular complexity: the role of relatedness and environmental constraints" has been accepted for publication in Proceedings B.

You can expect to receive a proof of your article from our Production office in due course, please check your spam filter if you do not receive it. PLEASE NOTE: you will be given the exact page

length of your paper which may be different from the estimation from Editorial and you may be asked to reduce your paper if it goes over the 10 page limit.

Your article has been estimated as being 9 pages long. Our Production Office will be able to confirm the exact length at proof stage.

Open Access

Paper charges

Sincerely,

Author's Response to Decision Letter for (RSPB-2019-2963.R0)

See Appendix B.

RSPB-2019-2963.R1 (Revision)

Review form: Reviewer 1

Recommendation

Accept with minor revision (please list in comments)

Scientific importance: Is the manuscript an original and important contribution to its field?

Good

General interest: Is the paper of sufficient general interest?

Good

Quality of the paper: Is the overall quality of the paper suitable?

Good

Is the length of the paper justified?

Yes

Should the paper be seen by a specialist statistical reviewer?

No

Do you have any concerns about statistical analyses in this paper? If so, please specify them explicitly in your report.

No

It is a condition of publication that authors make their supporting data, code and materials available - either as supplementary material or hosted in an external repository. Please rate, if applicable, the supporting data on the following criteria.

Is it accessible?

Yes

Is it clear?

Yes

Is it adequate?

Yes

Do you have any ethical concerns with this paper?

No

Comments to the Author

The authors have substantially revised the manuscript. It seems that the statistical analyses reveal a trend that is inherent to the data and not imposed by the analyses, which was a major concern raised in the first round of reviews. However, we still feel that some of the problems associated with the classification of cell types in the dataset were not sufficiently addressed. In particular, we would like to see a bit more discussion on how the different authors of the datasets used in this paper classified cell types across different multicellular lineages, and how this affects the results in this paper. This is important because the classification in itself could have implicit biases in the number of cell types counted, especially because a substantial portion of the data is now fairly old (i.e., Bell 1997), and modern molecular techniques have changed our views of what counts as a cell type. Otherwise, the paper is nice!

Review form: Reviewer 2

Recommendation

Accept as is

Scientific importance: Is the manuscript an original and important contribution to its field?

Good

General interest: Is the paper of sufficient general interest?

Excellent

Quality of the paper: Is the overall quality of the paper suitable?

Good

Is the length of the paper justified?

Yes

Should the paper be seen by a specialist statistical reviewer?

No

Do you have any concerns about statistical analyses in this paper? If so, please specify them explicitly in your report.

No

It is a condition of publication that authors make their supporting data, code and materials available - either as supplementary material or hosted in an external repository. Please rate, if applicable, the supporting data on the following criteria.

Is it accessible?

Yes

Is it clear?

Yes

Is it adequate?

Yes

Do you have any ethical concerns with this paper?

No

Comments to the Author

The authors adequately addressed the referees' concerns in their revision. In particular, they clarified the statistical methods used, which answered my main concerns. As such, I believe the manuscript is suitable for publication in the current form.

As a note, in line 265 the authors refer to "prokaryotes and archaea". I think they mean "bacteria and archaea" (or otherwise simply "prokaryotes") since archaea are prokaryotes also.

Appendix A

Response to Reviewers comments

Associate Editor

Comments to Author:

All three peer reviewers agree that the paper makes an important contribution to the understanding of the transition to multicellularity, particularly in its incorporation of scaling relationships.

However, this manuscript requires revision to be appropriate for publication in Proc B. Specifically, multiple reviewers have concerns about the selection of the breakpoint between "large" and "small" organisms, and the statistical support for a model that estimates different power law regimes for each of these categories, as opposed to a single unified model. Please address this concern as well as the other comments and suggestions made by the individual reviewers. Thank you,

We are grateful for the chance to submit a revised version of this manuscript in which we carefully responded to the helpful comments from reviewers. On advice from the reviewers, we have improved the following aspects of the manuscript:

- 1. The statistical justification for the scaling breakpoint analysis, and 2) clarified statistical support for the statistical models regarding power laws.*
- 2. We have made our data more visible, by improving the phylogeny and including it in the main text of the manuscript.*
- 3. We have also taken on board multiple comments about how the fungi are included in our study, dedicating a full paragraph (lines 286 – 304) to describe how they are particularly unique.*
- 4. We have also made more explicit the limitations of the data we use in the Materials and Methods (lines 316 – 327) and in the Discussion (lines 251 – 268).*

Reviewer(s)' Comments to Author:

Referee: 1

Comments to the Author(s)

The manuscript probes a previously published data set for a series of questions regarding multicellularity evolution. This data set spans a phylogenetically diverse set of eukaryotes, representative of the majority of known eukaryotic multicellular lineages.

Firstly, the authors use scaling theory to quantify the association between the organism size (measured by the mean number of cells of an organism) and its complexity. As a proxy for complexity, they rely on the number of cell types in each species. They conclude that the relationship between the number of cell types and the total number of cells in an organism follows two different regimes: one for small and another for large organisms.

The second question analyzed is the effect of the environment of origin in the process of group formation (clonal or non-clonal). Their phylogenetically informed approach corroborates Bonner's observation that clonal group formation is more common when the transition happens in an aquatic environment.

Finally, the authors look into the complexity level of multicellular organisms on land vs in
water. They find that the complexity is in general higher for land organisms, a result that
remains strong even after controlling for phylogenetic factors.

The manuscript is clear and engaging to read, and the issue investigated is relevant. I
particularly like the strong statistical confirmation that clonal group formation is more
frequent for aquatic transitions. Nevertheless, I have some concerns regarding the scaling
relations obtained.

**Major comments**

The authors conclude that two different power-law regimes exist, corresponding to small
and large organisms. However, in my view, the evidence presented does not support this
conclusion. The R^2 for the fitting to the full data is larger than the ones obtained for partial
data in each subset, suggesting that a fit to the full data retains a relatively high explanatory
power without requiring additional hypotheses.

Furthermore, the authors state that they used an R package to test for the existence of the
breakpoint, but provide no test result.

*Visual inspection of our regression suggested that a linear ordinary least square regression*
*did not adequately describe the scaling dynamics based on our dataset. Here is a*
*representation of this non-linear 'breakpoint' with a smoothing curve:*

*We first note that the R^2 using a single regression for the full dataset (0.639) is actually*
*very similar to the R^2 for the lower break (0.61) but is actually very different than R^2 for the*
*upper break (0.15). Combined with the statistical support for retaining the breakpoint*
*analysis (see below), we feel that the two sides of the break represent different scaling*
*dynamics and that we gain important biological insights by retaining this approach.*

*To evaluate our visual conclusion, we used the segmented package to statistically estimate*
*whether a breakpoint is justified and objectively determine the placement of the breakpoint,*

as is commonly applied in the literature. This procedure uses maximum likelihood to
estimate a γ term that measures the distance between the end of the first segment and the
beginning of the next. The model converges when the γ term is minimized. According to this
procedure, we are required to supply a 'best guess estimate' of the breakpoint (psi). We
tried a range of reasonable values from $X = 4$ to $X = 10$, and the analysis always yielded a
breakpoint of 4.8 (SE: 0.898).

**Our R code:**

`model <- glm(log10celltypes ~ log10totalcells, data = complexity1)`
`segmodel <- segmented(model, seg.Z = ~log10totalcells, psi = 8)`

**where psi = a value from 4 to 10*
**Seg.Z = is a one sided formula describing the predictor with a segment (we only have one*
*predictor, x, which has the segment)*

**This segmented analysis yielded the output:**

****Regression Model with Segmented Relationship(s)****

*Call:*
`segmented.glm(obj = model, seg.Z = ~log10totalcells, psi = 8)`

*Estimated Break-Point(s):*
*Est. St.Err*
***psi1.log10totalcells 4.8 0.898***

*Meaningful coefficients of the linear terms:*
*Estimate Std. Error t value Pr(>|t|)*
*(Intercept) -0.23753 0.09866 -2.408 0.0176 **
*log10totalcells 0.21439 0.03472 6.175 8.98e-09 ****
*U1.log10totalcells -0.14193 0.03883 -3.656 NA*

---
*Signif. codes: 0 '***' 0.001 '**' 0.01 '*' 0.05 '.' 0.1 ' ' 1*
*(Dispersion parameter for gaussian family taken to be 0.1033765)*

*Null deviance: 39.075 on 125 degrees of freedom*
*Residual deviance: 12.612 on 122 degrees of freedom*
*AIC: 77.566*

*Convergence attained in 0 iter. (rel. change 8.5323e-09)*

*See the reference ([#https://www.r-bloggers.com/r-for-ecologists-putting-together-a-piecewise-regression/](https://www.r-bloggers.com/r-for-ecologists-putting-together-a-piecewise-regression/)) for details about the method and our analytical approach.*

**Minor comments**

- line 113-114: the value of R^2 from OLS is reported amidst the values of RMA fitting,
which may be misleading

*Based on this reviewer's comments and referral to the Smith (2009) paper, we have*
*removed the RMA analysis from the manuscript and instead use OLS.*
- line 375: is table S1 the correct reference?
*Correct references inserted.*
- table S2: N is 126 for the small species. Is it not 50? *Corrected.*
- table S2: no R² provided. *We have removed the RMA analyses from the paper (see*
*comment below).*
- tables S3-5: no N provided. *Corrected.*
- the authors use reduced major axis regression with the argument that both X and Y
variables have associated errors. It is not clear to me that this is the best approach to take
(please see, e.g., RJ Smith, Use and misuse of the reduced major axis for line-fitting, 2009)
*We have carefully read Smith (2009) who argues that the decision to use OLS vs RMA is*
*not simple and depends on a biologist's view of the data. Moreover, while RMA is frequently*
*used by biologists when the X axis variable is measured with systematic error, Smith*
*argues it is actually more important to consider symmetry relationship between the X-Y*
*variables. Specifically, OLS should be used if the X and Y variables cannot be reversed (i.e.*
*asymmetry). Smith goes on to divide sources of error between 'measurement' and 'natural'*
*variation, where measurement error combines technical error (e.g. low precision when*
*estimating cell number) + sampling error (e.g. available data in the sample may not*
*adequately represent the total 'population' of species). Natural variation (e.g. age and sex-*
*dependent variation) is considered to be a distinct source of error since this only exists in*
*the context of the specific regression analysis (i.e., there is no 'true' value of the variable, as*
*there is for sources of measurement error). Thus, natural error should not be used to decide*
*whether or not to use OLS or RMA.*
*In the end, we used Smith's (2009) checklist and the symmetry criterion to decide to*
*remove the RMA analysis from the manuscript and use OLS as suggested by the reviewer:*
*1) X 'causes' Y*
*2) X in some noncausal manner restricts, limits, or determines Y*
*3) X is being used to predict values of Y*
*4) The purpose of the regression is to understand the range of values Y can take at any*
*given value of X*
*5) Change in Y is a response by correlated evolution to selection on X*
- I would like to see a comment on why the authors have only considered eukaryotes in this
study, given that the original data set (Fisher et al., 2013) included species from all three
domains of life.
*We have added the following explanation to the text: "We focus only on the eukaryotic*
*species included in the Fisher et al. (2013) dataset, as it is unfeasible to apply the same*
*standards of allocating either 'aquatic' or 'terrestrial' to prokaryotes and archaea".*
*Explanation included in Materials and Methods (line 321 – 323).*

**Referee: 2**

**Comments to the Author(s)**

Fisher et al test an old hypothesis by Bonner that postulates that the environment
determines the type and stability of multicellular associations that emerge. I found the paper
very interesting, concise and the analyses robust. I have minor suggestions to the authors
that they may take or leave.

The aspect that made me think the most is how reliable the input data are that the authors
took from Bell and Mooers 1997? Looking at the table in this publication, I have the feeling
that its both taxonomically biased and inaccurate in many cases. I'm not questioning Bell-
Mooers's efforts in compiling those data, but state of the art at the time might not have
allowed them to reach high accuracy in many cases. I can only assess fungi, where I feel
serious limitations. I'm not sure what the authos could do to update these figures or to
mitigate their potential bias on their results, but at least some discussion would be useful.
*The dataset we use in this study is from Fisher et al (2013), which is largely comprised of the*
*Bell & Mooers (1997) dataset, however also includes species from a broader literature*
*search. The dataset of Bell & Mooers (1997) that we use for information on the number of*
*cell types and the total number of cells, whilst a fantastic resource for these types of studies,*
*has its own drawbacks. A major concern is that the data is not up-to-date and is biased*
*towards particular taxa. We acknowledge this concern and agree that the data is not always*
*current. However, it is worth bearing in mind that for a species to be included in our full*
*analysis, where we include data on complexity (Table S5 - number of cells types and total*
*number of cells as a response variable), which is the main way that we use the Bell & Mooers*
*data, we must have data on both the number of cell types but also on the total number of*
*cells. This makes the Bell & Mooers (1997) dataset quite unique in that it provides information*
*for both these traits.*

*We have made this limitation clearer in the Materials and Methods (lines 316 - 318) and in*
*the Discussion (lines 251 – 268).*

As a general suggestion I think the data used in the analyses could be made more
accessible by including it in the main text. Consider moving Figure S1 to the main text and
displaying the examined traits next to taxon names (like in a circos plot).

*We have edited and moved the phylogeny – is now Figure 4. It now includes more*
*information than the previous phylogeny in the Supplementary Information, including the*
*current environment and the number of cell types.*

**Specific comments:**

I34 - wild type yeast under normal condition does not form aggregates (it forms biofilms, but
thats not an aggregation).

*Here we are referring to adhesion through mannose residues found in Saccharomyces*
*cerevisiae, mediated by the flocculins FLO1, FLO5, FLO9 and FLO10. We describe this*
*type of interaction as aggregation because it is non-clonal group formation (i.e. genetically*
*distinct cells that are able to adhere and form a multicellular group).*

I55 - as a mycologist I argue that ref10 has outdated numbers for fungal cell type diversity,
which can reach 30 (which is based only on simple morphological observations, Kues 2015
Fungal Biol Rev., Bistis & Read 2003 Fungal Genet Biol.)

*We thank the reviewer for the updated references. We have included a Discussion*
*paragraph giving more credit to the Fungi (lines 286 – 304).*

*We acknowledge the extra references given here. However, in order for them to be*
*included in our complexity analyses, we need to have data for the number of cell types,*
*total number of cells, current environment, mode of group formation, and ancestral*
*environment. Moreover, including these extra species in the phylogenetic analyses of*
*independent transitions to multicellularity would not change the results, only add minimal*
*power to the analyses, as we are interested in whole lineages and not numbers of species.*
I93-95 - the sentence reads like it has a bit of circularity in logic.
*This sentence has been removed.*
I234 - As an alternative explanation, I would also argue that water currents distribute
nutrients to all the cells of a clonal colony (in a sponge for example), whereas goods are
more patchy on land and therefore need to be actively foraged for.
*We agree that this could be another reason why non-clonal group formation might be more*
*common on land. However, we have not included this additional hypothesis in the main*
*text, as there could be many non-competing explanations.*
I237 - please mention the containing lineages not example genera
*Corrected.*
I239 - yes, but fungi with motile cells are connected to water. I would argue that multicellular
fungi don't have a motile cell type, because fast apical growth of hyphae enables foraging
for nutrients.
*Have re-phrased the sentence.*
I246 - Kiss et al (Nat Comms 2019) argues in a different way for facultative multicellularity in
yeasts, the conservation of multicellularity-related genes and facultative hypha formation.
*We include a discussion of this in the Discussion (lines 297 – 301).*
I250 - I don't see a direct link between being obligate and being big. Please rephrase.
*There is some evidence that there is a link between being big and being obligate – Fisher et*
*al 2013 show that obligate species have significantly more cells than facultative species*
*(Figure S3d, in Supp Info, $p = 0.02$) but only when phylogenetic relationships are not taken*
*into account. We have rephrased to reflect this uncertainty (lines 259 – 261).*
I270 - there is some discussion on that in the below mentioned papers and in Nagy et al
Microb Spectrum 2017. "Hyphal multicellularity" indeed seems to be an exception from
many rules, and its not clearly clonal in the classic sense (clonal et the level of nuclei, not at
cells).
*We have included a more in depth discussion of how fungal multicellularity compares to*
*other lineages in the Discussion (lines 286 – 304).*
I279 - possibly? *Have rephrased to 'There are also other potentially confounding*
*physiological parameters, for example the possibility that autotrophic lineages compete for*
*light, both on land and in the water, whereas heterotrophic lineages do not.'*
I301 - finishing the discussion with a broad conclusion on multicellularity would probably

read better to me than a somewhat forced generalization to completely unrelated
organismal traits.
*Have altered the final discussion paragraph (lines 306 - 310).*
I322 - where was data for the ancestral environment of lineages taken? Ideally, this attribute
could be inferred by ancestral state reconstructions.
*References for ancestral environments are included in Table 1. These are general*
*references and review papers that discuss the ancestral environments of the lineages.*
*Ancestral state reconstructions would have been possible, but are only really useful when*
*the ancestral state is unknown.*

I341 - see Kiss et al Nat Comms 2019 and Nagy 2018 Biol Reviews on an update on fungi.
The transition to simple multicellularity likely occurred once, whereas complex
multicellularity emerged several times.
*We have included a more in-depth discussion of how fungal multicellularity compares to*
*other lineages in the Discussion (lines 286 – 304). We have additionally made multiple*
*minor adjustments throughout the paper to include this reasoning and other similar*
*comments.*

Laszlo G. Nagy

**Referee: 3**

**Comments to the Author(s)**

Report on “The evolution of multicellular complexity: the role of relatedness and
environmental constraints” by RM Fisher et al.

The paper has a compelling history, a fair amount of data analysis and metanalysis. The
authors test the hypothesis (by John T. Bonner) that the evolution of multicellularity is
affected by where such transition occur: in an aquatic vs. terrestrial environment. In
particular, what is interesting is that clonality vs. aggregativity differ significantly in the
number of cell types (and organismal size), which could be well explained by how the
environment favored one or the other type of transition to multicellularity. By doing an
extensive meta-analysis of the literature, the authors concluded that the theory of social
evolution (relatedness, in particular) cannot completely account for these differences, and
that the environment has played a major role. I believe that the paper makes an important
contribution to the understanding of the transition to multicellularity, and it also brings forth a
novel perspective in scaling relationships that (so far) have been seldom studied in the
literature on multicellular evolution.

I would suggest the following discussion/clarification points:

1. Why is the isometric scaling the frame of reference (the null hypothesis)? It seems
unrealistic that any organism would ever evolve one cell type directly proportional to its cell
count. Thus, a null model that cannot ever exist(?), even in principle, might not be that
useful.

*We can see the reviewer’s point here that isometry isn’t a ‘null model’ and have rephrased*
*this passage (lines 76 – 80) to read:*

*Isometric scaling ($b = 1$) provides a (likely unattainable) theoretical upper limit on cell*
*diversification rates, since cell type and cell number would increase at the same rate (i.e.*
*every added cell is a new type). Below this upper limit, allometric scaling ($b < 1$) would*
*indicate that cell type increases at a slower rate than cell number (i.e. small organisms have*
*more cell types relative to their body size).*
2. Is there a conflict in the way different authors in the papers cited conceive (and account
for) a “cell type”? If so, how is this addressed in a way that makes the conclusions robust to
such discrepancies?
*Yes, different authors define cell types differently and we have included a brief discussion*
*of this problem in the Discussion (lines 265 - 267).*
3. Considering the wide range of organisms studied, it also seems a sensible discussion to
mention how the developmental stage at which the cell types and cell number were
“counted” could affect the conclusions. Are these cell types the total number of types
present throughout the life cycle? Are they the types at a “fully developed” organism? How
these temporal/developmental boundaries can be compared across such diverse groups of
organisms?
*We have included a brief discussion of this problem in the Discussion (lines 265 - 267).*
4. I understand that one can break a regression into multiple sections, but why was 2
groupings the chosen one? (at 10^4 cells). Even though it gives a ‘nice’ discussion
between “small” vs. “big” organisms, it seems a bit arbitrary. Was it sensitive to the
sampling distribution (the small organisms seem to be “overrepresented” at ~ 4 cell types)?
At the boundary, are there organisms qualitatively distinct? (i.e. close taxa grouped in one
or the other groups).
*Please see our response to Reviewer 1, where we provide a detailed overview of the*
*statistical approach (segmented package in R) used to objectively identify the ‘breakpoint’*
*in the data. In this way, the breakpoint of 4.8 (6.3×10^4) cells is not arbitrary, but rather it*
*arises objectively from our dataset. If we change the dataset by removing many of the*
*smallest taxa, the breakpoint is likely to change—but, it would also change if we removed*
*many of the largest taxa. Yet, we also appreciate the reviewer’s point about using this result*
*to refer to discrete classes of ‘small’ and ‘big’ organisms. Thus, we have removed this*
*framing of the breakpoint result from the entire manuscript and now refer instead to a*
*transition along a continuum of body size where scaling constraints appear to change.*
5. Even though there is a discussion of sampling bias at the end, could the authors test
the robustness of the main conclusions to sampling biases? (i.e. some bootstrap method on
the datasets, weighted perhaps by taxa?).
*Our analyses are phylogenetically-controlled, meaning that shared ancestry is taken into*
*account. Therefore, sampling bias within an independent lineage (i.e. more animal species*
*than plant species) will have limited impact on the results of our phylogenetic analyses, as*
*these use contrasts between independently evolved lineages. In other words, increasing*
*the number of species in our dataset within a particular lineage does not increase the*

*sample size of the analysis – only increasing the number of independent contrasts between*
*lineages will increase the sample size. We are confident that we have a fair sampling of*
*multicellular lineages (only missing Stramenophiles and the diatoms).*

6. Some of the data, especially on the number of independent transitions to multicellularity,
is not current with the latest phylogenetic developments. Specifically, the 'red algae' are
described as having 1+ transitions to multicellularity in Figure 1 (citing Andy Knoll's 2011
review paper)- these two papers say either 2 or 3 times:

Cock JM, Collén J (2015) Independent emergence of complex multicellularity in the brown
and red algae in Evolutionary Transitions to Multicellular Life. (Springer), pp. 335–361.

Yoon HS, Müller KM, Sheath RG, Ott FD, Bhattacharya D (2006) Defining the major
lineages of red algae (rhodophyta). Journal of phycology 42(2):482–492.

Perhaps more importantly, the fungi have recently come into focus much more than before.
The old estimate was 2 origins, but Lazlo Nagy's groundbreaking 2018 paper showed it was
likely between 8 and 11.

Nagy LG, Kovács GM, Krizsán K (2018) Complex multicellularity in fungi: evolutionary
convergence, single origin, or both? Biological Reviews 93(4):1778–1794

*We have been conservative with our estimates of how many times multicellularity has*
*evolved in each lineage, to reflect some uncertainty in these estimates and to avoid over-*
*counting in more well-studied lineages. Therefore, we have entered, for example, '1+*
*transitions' for the red algae, to acknowledge that there were most likely multiple transitions*
*in this group whilst also simplifying our analyses and assumptions.*

*As we understand it, the estimates in Lazlo Nagy's 2018 paper are for the number of times*
*complex multicellularity has evolved within the Fungi. We focus on multicellularity in*
*general, whether it be simple or complex (which in themselves are hard to consistently*
*define across taxa).*

**Small observations:**

1. In Fig. 1, the X-axis legend in the left panel reads "Total number of cells" and the right
panel reads "Total number of cells (log)". I'd keep the legends consistent.

2. X-axis legend of Fig. 3 (right panel) seems to have formatting errors (with 10^{11} and

*Have corrected these formatting errors.*

**Tracked changes from original – resubmitted version**

**The evolution of multicellular complexity: the role of relatedness and**

430

**environmental constraints**

Fisher, RM¹, Shik, JZ^{1,2} & Boomsma, JJ¹

¹Section for Ecology and Evolution, Department of Biology, University of Copenhagen,
Denmark

²Smithsonian Tropical Research Institute, Apartado 0843-03092, Balboa, Ancon, Republic
of Panama

**Keywords:** major evolutionary transitions, multicellularity, scaling

**Abstract**

A major challenge in evolutionary biology has been to explain the variation in multicellularity
across the many independently evolved multicellular lineages, from slime moulds to

~~humans~~vertebrates. Social evolution theory has highlighted the key role of relatedness in
determining multicellular complexity and obligateness, however there is a need to extend

this to a broader perspective incorporating the role of the environment. In this paper, we
formally test Bonner's 1998 hypothesis that the environment is crucial in determining the

course of multicellular evolution, with aggregative multicellularity evolving more frequently
on land and clonal multicellularity more frequently in water. Using a combination of scaling

theory and phylogenetic comparative analyses, we describe multicellular organisational
complexity across 139 species spanning 14 independent transitions to multicellularity and

investigate the role of the environment in determining multicellular group formation and in
imposing constraints on multicellular evolution. Our results, showing that the physical

environment has impacted the way in which multicellular groups form, ~~could shed light on~~
~~the role of the environment for other~~highlight that environmental conditions might also have

affected the major evolutionary transitions.

**Introduction**

Macroscopic life on earth has been shaped by the evolution of multicellularity from unicellular
ancestors. Multicellularity ~~is a complex and variable trait, ranging~~ranges from simple cell

aggregations found in yeast to differentiated metazoan organisms, with much diversity in
between [1]. For example, our bodies contain 10^{14} cells with more than 200 specialized types
[2] but *Volvox* is 10 orders of magnitude smaller and has just 2 cell types [3]. Some lineages
have become obligately multicellular, where cells only exist as part of a multicellular organism
(e.g. animals), whereas others remain facultative, switching between a unicellular and
multicellular lifestyle (e.g. cellular slime moulds) [4].

A major challenge in evolutionary biology has been to explain this variation in complexity
among multicellular lineages. Social evolution theory has greatly advanced our
understanding of the evolution of multicellularity, primarily through clarifying the factors that
favour the cooperation needed to become multicellular. We understand how relatedness
between cells is crucial in determining when altruism can evolve ~~(, as for example,~~ in
*Dictyostelium* slime moulds) [5], where division of labour between cell types [6] and the
proliferation of cheaters [7-9], have been amply studied. It has also become clear that clonal
relatedness ($r = 1$) is a necessary, albeit not sufficient, condition for the evolution of obligate
multicellularity like we see in animals and plants, and that these lineages have more cell
types than those with facultative multicellularity [4].

However, there are limits to the variation in multicellular complexity that is explained by
relatedness. For example, both land plants and fungi have cells that are clonal and obligately
multicellular, but plants have approximately 10 times more cell types than fungi [10] and it is
unclear what can explain these differences. There are good reasons to speculate that the
environment could be an important factor shaping the first trajectories of multicellular
evolution with lasting consequences for later elaborations. Firstly, the environment itself could
determine the way in which multicellular groups form and hence relatedness between cells.
Bonner (1998) observed that clonal group formation, where daughter cells remain attached
to mother cells after division, seems to be more common in lineages that originated in the
sea compared to species that originated on land [11]. If this is the case, it would mean that
the environment where multicellularity originates could have profound consequences for
subsequent evolutionary possibilities. Secondly, the physical constraints associated with
living in water or on land are likely to affect many aspects of phenotypic evolution, for example
the need for support and structural reinforcement tissues, the diversity of dispersal
mechanisms, and ~~in-sustaining~~ the biomechanics needed to sustain active motility.

Scaling theories provide powerful tools to test for such constraints, since an organism's
body size can accurately predict complex traits such as metabolic rate, lifespan, and growth
rate [12], and since the shapes of these relationships reflect fundamental physiological
constraints on how diverse organisms can evolve [13]. Scaling relationships can also reveal
outlier taxa ~~that highlight cases~~ where evolutionary innovation fueled the breaking of
ecological and physiological constraints [14]. In practice, scaling parameters (*i.e.* the slope
(*b*) and intercept (*a*) in the equation $y = aM^b$) represent mechanistic hypotheses that, for the
purpose of this study, relate the number of cell types (*y*) to the total number of cells (*M*).

Isometric scaling ($b = 1$) provides a ~~null model, predicting that (likely unattainable)~~
~~theoretical upper limit on cell diversification rates, since~~ cell type and cell number ~~would~~
~~increase at the same rate (i.e. every added cell is a new type), and). Below this upper limit,~~
allometric scaling ($b < 1$) would indicate that cell type increases at a slower rate than cell
number, ~~such that (i.e. small organisms have more cell types relative to their body size).~~

There is a need to build on our understanding of the fundamental factors influencing
multicellular evolution – primarily the role of relatedness – and extend this to a broader
perspective incorporating the role of the environment. The objectives of this paper are to: (1)
describe the variation in multicellular organisational complexity across 139 species by
investigating the scaling relationships between body size (total number of cells) and number
of cell types; (2) use phylogenetically-controlled comparative analyses across 14
independent multicellular transitions to assess the extent to which the environment
determines how multicellular groups form and the consequences for whether obligate
multicellularity ~~evolves could evolve~~; and (3) test whether constraints imposed by the
environment can explain why some lineages have reached higher levels of organisational
complexity than others, and can account for part of the variation in cell-type diversity and
differences in scaling relationships. ~~We use the term organisational complexity to highlight~~
~~that division of labour is fundamental to multicellularity, that the number of cell types is a~~
~~marker of division of labour and that any form of division of labour requires organisational~~
~~integration.~~

Results

Describing variation in body size and complexity

Formatted: Left

Formatted: Font color: Custom Color(34,34,34)

Formatted: Font: Not Italic, Font color: Custom Color(34,34,34)

Formatted: Font color: Custom Color(34,34,34)

Formatted: Font color: Custom Color(34,34,34)

Formatted: Font color: Custom Color(34,34,34)

Formatted: Font color: Custom Color(34,34,34)

Formatted: Font color: Custom Color(34,34,34)

Formatted: Font: Not Italic, Font color: Custom Color(34,34,34)

Formatted: Font color: Custom Color(34,34,34)

Formatted: Font color: Custom Color(34,34,34)

Formatted: Font color: Custom Color(34,34,34)

Commented [RF1]: I think it best to just remove this sentence?

Across the species in our dataset, representing 14 independent transitions to multicellularity,
we found that the scaling of cell type and cell number is strongly allometric (~~reduced major~~
~~axis~~ordinary least squares regression, ~~RMA~~ OLS): slope = 0.1411, CI = 0.1310 - 0.1613, R^2
= 0.64, Figure 1a & b). This means that despite a positive association, the number of cell
types increases much more slowly with cell number than arithmetic proportionality (i.e.
isometric scaling) would predict. In other words, small organisms are organisationally more
complex for their size than large organisms.

We next found evidence of a transition along the continuum of body size where the scaling
relationship between number of cell types and total number of cells changed (Table S1, S2).
We used a statistical approach to estimate this breakpoint in the regression at 6.3×10^4 total
number of cells, corresponding to approximately 6 cell types. Smaller species (before the
breakpoint) showed an allometric slope about three times as steep (OLS: slope = 0.21, CI =
0.16 – 0.26, $R^2 = 0.61$) as larger species (above the breakpoint) (OLS: slope = 0.07, CI =
0.03 – 0.11, $R^2 = 0.15$). This opens up the possibility that organisms face additional
constraints in the accumulation of new cell types as they evolve larger body sizes, and
supports the observation that lineages consisting of small species gain new cell types more
quickly as they grow in size.

~~We next found that there was a difference in the scaling relationship between number of cell~~
~~types and total number of cells for small versus large species (Table S1, S2). Specifically,~~
~~we identified the estimated breakpoint in the regression as 6.3×10^4 total number of cells,~~
~~corresponding to 4.8 ± 0.9 cell types, where the scaling relationship changes. Small species~~
~~(before the breakpoint) showed an allometric slope about twice as steep (RMA: slope = 0.27,~~
~~CI = 0.23 – 0.33, $R^2 = 0.61$) as large species (above the breakpoint) (RMA: slope = 0.18, CI~~
~~= 0.15 – 0.23, $R^2 = 0.15$). This implies that larger organisms face a unique set of more~~
~~stringent constraints on the accumulation of new cell types than small organisms, and~~
~~supports the observation that lineages consisting of small species gain new cell types more~~
~~quickly as they grow in size.~~

We further sought to understand the substantial reduction in cell type variation explained by
cell number for large organisms (i.e. $R^2 = 0.15$) using a technique called quantile regression.
Regression through the upper 90% quantile of the dataset suggests that there is an upper
threshold to the number of cell types a species can have for its size, whereas there is ~~a lot~~
~~of~~substantial variation in the number of cell types below that threshold (Figure 1a, dashed

Formatted: Font color: Text 1

Formatted: Font color: Text 1

Formatted: Font color: Text 1

Formatted: Font color: Text 1

Formatted: Font color: Text 1

Formatted: Font color: Text 1

Formatted: Font color: Text 1

Formatted: Font color: Text 1

Formatted: Font color: Text 1

Formatted: Font color: Text 1

Formatted: Font color: Text 1

Formatted: Font color: Text 1

Formatted: Font color: Text 1

Formatted: Font color: Text 1

Formatted: Font color: Text 1

Formatted: Font color: Text 1

Formatted: Font color: Text 1

lines). This suggests that: there could be other factors limiting the number of cell types below
 that threshold and these other limiting factors are especially important in larger species since
 the slope describing this upper limit is ~~far shallower~~ about half as steep ($b = 0.13$) than the
 upper limit for small species ($b = 0.25$) (Table S2).

**Figure 1(a) Scaling across multicellular organisms.** The relationship between number of cell types
 and total number of cells for ~~small~~ smaller (in black, ~~between 4–10⁴ cells~~) and ~~large~~ larger multicellular
 species (in grey, ~~between 10⁴–10¹⁴ cells~~) shown on logarithmic axes. Small species show a steeper
 allometry (~~reduced major axis OLS regression: slope = 0.2721 (CI = 0.23–16 – 0.3326)~~) compared to
 large species (~~reduced major axis OLS regression: slope = 0.1807 (CI = 0.1503 – 0.2311)~~). Solid lines
 show the ~~reduced major axis OLS regression~~ and dashed lines show regressions through the upper

Formatted: Font color: Text 1
 Formatted: Font color: Text 1

90% quantile of the data. We estimated the breakpoint of 4.8 ± 0.9 (corresponding to 6.3×10^4 total cells) using the 'segmented' package in R. **(b) Multicellular organisational complexity across different multicellular lineages.** Organisational complexity, measured as both the number of cell types and the total number of cells, for each of the independently evolved multicellular lineages. Original data are from Fisher *et al* (2013) and images of *Mus musculus* and *Volvox* are from Phylopic (<http://phylopic.org/>). The statistical results of the different regressions are given in Table S1. N total = 126 species where we had data for both number of cell types and total number of cells.

The origins of multicellularity in different environments

Our results show that the physical environment (whether or not ~~a species lives~~ ancestral lineages lived in the water or on land) has had a major impact on both the origins and subsequent elaborations of multicellularity, both in determining how multicellular groups originally ~~form~~ formed and how organisational complexity subsequently ~~evolve~~ evolved.

We found that lineages in aquatic environments were significantly more likely to form multicellular groups through daughter cells remaining attached to mother cells after division (clonal group formation) (MCMCglmm, difference between aquatic & terrestrial: posterior mode = 5.74, credible intervals (CI) = 2.91 – 9.79, $p_{diff} = 0.0008$, $N_{species} = 139$, Figure 2a). All of the multicellular lineages in our dataset that have their origins in water form multicellular groups in this way, whereas two thirds of the lineages that originated on land form groups through aggregation (non-clonal group formation) (Figure 2a). In fact, there are only two lineages ~~that~~ where multicellularity originated on land that employ clonal group formation – the Fungi and the plasmodial slime moulds and these tend to grow in terrestrial environments of saturated humidity. This result confirms Bonner's original observation that clonal group formation is more common in multicellular lineages originating in the sea [11].

Secondly, we found that the transition to obligate multicellularity was significantly more likely to occur in aquatic environments compared to on land. Most (5 of 6) lineages that evolved multicellularity on land remained facultatively multicellular (difference between aquatic & terrestrial: posterior mode = 6.59, CI = 4.29 – 8.72, $p_{diff} < 0.0001$, $N_{species} = 139$, Figure 2b). The only multicellular lineage that has evolved obligate multicellularity on land is the Fungi. This is consistent with this lineage also being a rare example of clonal group formation that originated on land, as the resulting clonal relatedness between cells is significantly associated with the transition to obligate multicellularity [4].

Table 1: At least 14 transitions to multicellularity occurred within the eukaryotes. Estimates of the number of independent transitions to multicellularity in each lineage are given along with the environment where the lineage originated when it evolved multicellularity, average number of cell types, and the corresponding references. We have not included two other known transitions to multicellularity – the diatoms [15] and *Sorodiplophrys* (Stramenopiles) [16] – due to a lack of data on cell types and environment of origin. See Figure 3a3. *The Fungi evolved obligate multicellularity on land twice - once in the Ascomycota and once in the Basidiomycota (Knoll 2011, Stajich et al 2009). However, we only have examples of Ascomycota in our dataset, so write this as '1+' here, to avoid inconsistency with the number of transitions in Figure 2.

Lineage	NumberEs timated number of transitions	Average number of cell types	Ancestral environment where multicellularit y evolved	Obligate or facultative multicellularit y	Reference(s)
Acrasid slime moulds	1	2	terrestrial	facultative	[17]
Brown algae	1	6.9	aquatic	obligate	[18,19]
Cellular slime moulds	1	2	terrestrial	facultative	[20]
Chlorophyte algae	1 - 4	1.5	aquatic	facultative & obligate	[21]
Choanoflagellates	1	1	aquatic	facultative	[22]
Ciliates	2	2.5	terrestrial & aquatic	facultative & obligate	[23-25]
Fonticula alba (Fonticulida)	1	2	terrestrial	facultative	[26]
Fungi	21+*	7	terrestrial	facultative & obligate	[27,28]
Metazoa	1	101.6	aquatic	obligate	[28]
Oomycetes	1	1	aquatic	facultative & obligate	[29]
Plants	1	22.2	aquatic	obligate	[28]
Plasmodial slime moulds	1	2	terrestrial	facultative	[30]
Red algae	1+	10.8	aquatic	obligate	[28]

Commented [RF2]: Update here + new reference

Commented [RF3]: Update here + new reference

623

624

Figure 2: The origins of multicellularity in different environments. (a) The proportion of lineages that have clonal group formation that originated in aquatic and terrestrial environments. All multicellular lineages that originated in the sea have clonal group formation (8/8 lineages) whereas most of the multicellular lineages that originated on land have non-clonal group formation (4/6 lineages). **(b)** Multicellular lineages that originated in water more commonly evolve obligate multicellularity (6/8 lineages) compared to lineages that originated on land, which more often remain facultatively multicellular (5/6 lineages).

Multicellular organisational complexity on land versus in water

We found that the number of cell types of multicellular species currently found on land was significantly higher than those currently found in aquatic environments (Figure 3a3), whilst controlling for the total number of cells (posterior mode = -0.77, CI = -1.42 to -0.11, $p_{diff} = 0.02$, $N_{species} = 121$, Figure 3b4). The average number of cell types for aquatic lineages is 8 whereas for terrestrial lineages it is 25. Species on land were however not significantly larger in size than those found in the sea (posterior mode = -2.79, CI = -9.04 to 1.81, $p_{diff} = 0.12$, $N_{species} = 121$). Overall, there was a significant phylogenetic correlation between number of cell types and total number of cells, meaning that species with more cell types also tend to be bigger due to their shared ancestry (posterior mode = 0.90, CI = 0.72 to 0.96, $p_{diff} < 0.0001$, $N_{species} = 121$). However, we also found a significant phenotypic correlation between these two variables, meaning that the association is also a result of a shared environment (posterior mode = 0.56, CI = 0.19 to 0.76, $p_{diff} = 0.004$, $N_{species} = 121$).

Formatted: Font color: Text 1

649

650

651 **Figure 3: Organisational complexity and environmental constraints. a)** A summarised
 652 **phylogenetic** **Phylogeny** **of the multicellular lineages in our dataset.** Each lineage that have

Formatted: Font: 12 pt
 Formatted: Font: Bold
 Formatted: Font: Bold

independently evolved multicellularity. The within the eukaryotes is highlighted in a different colour – with two independent lineages within the Ciliates (N = 14). The number of cell types (log10) and the current environment is shown as of each species (blue = aquatic, green = terrestrial (in green), aquatic (in blue) or both (half green, half blue) for species that have a substantial number of species in both environments. b)) 
[revised manuscript text omitted]

~~to exclude some fungal data from the literature (Bistis et al 2003, X and X4)~~. Whilst the lack
of data is a problem for drawing detailed conclusions about scaling relationships of fungi, ~~this~~
~~would not affect~~ our phylogenetic analyses ~~(that focus on would be unaffected (as they deal~~
~~with independent contrasts between~~ lineages, not species ~~(Figure 2, Table 1))~~ *per se* so we
can be confident in our conclusion that clonality is associated with aquatic ancestral
environments (Figure 2a).

~~Secondly, it has been argued (Nagy et al 2018, Kiss et al 2019) that multicellular group~~
~~formation in the Fungi cannot be classified as clonal or aggregative.....strictly classified as~~
~~clonal or aggregative, due to the way in which multi-nucleate fungal hyphae form, and this is~~
~~a potential limitation of our study where we had to classify every species in these two discrete~~
~~categories. Finally, it is also difficult to confidently class fungi as either terrestrial or aquatic,~~
~~as they live and have evolved mostly at the air-water interface. Perhaps a closer look at the~~
~~Fungi as putative 'exceptions to the rule' could help to unravel the generality of the~~
~~relationship between the environment and multicellular complexity that we uncovered.~~
~~Finally, the it could also be argued that it is difficult to confidently class them as either~~
~~terrestrial or aquatic, as they live and have evolved at the air-water interface.~~

~~Perhaps a closer look at the Fungi as 'exceptions to the rule' could help to unravel the~~
~~relationship between the environment and multicellular complexity.~~

Commented [RF4]: Add reference and comment on Nagy et al 2017 – they mention this (check paper)

Our results, showing that the physical environment has impacted the way in which
multicellular groups form, could therefore shed light on the role of the environment for other
major evolutionary transitions (~~ref~~-~~West et al 2015, Boomsma & Gawne 2018~~) and help us
~~understand the balance between how intrinsic and extrinsic factors can affect evolutionary~~
~~trajectories.~~

**Material and Methods**

**Data collection**

The data used in this study were originally published in Fisher *et al.* (2013) and are stored in
the data depository Dryad (original data can be found here:
<https://datadryad.org/resource/doi:10.5061/dryad.27q59>). In summary, we conducted an
extensive literature search on multicellular species, searching specifically for information on
multicellular complexity (number of cell types and the total number of cells), the ways in which
groups formed and whether or not they were obligately or facultatively multicellular. We focus
only on the eukaryotic species included in the Fisher *et al.* (2013), as it is unfeasible to apply
the same standards of allocating either 'aquatic' or 'terrestrial' to prokaryotes and archaea.
Our full dataset can be found in Table S6. Information of both the number of cell types and
total number of cells (allowing us to estimate 'complexity') was essential for analyses where
we included complexity (Table S5) and we therefore focused our data collection effort on
species where information on both these traits was available.

In this study, we expanded on the eukaryotic species in the original dataset by adding
information on the ancestral and current environment of each species. We considered any
species found on land as terrestrial and any species found in freshwater, brackish or marine
environments as aquatic. We found information about the current environment of a species
by searching on Google Scholar for publications and also taxa-specific websites, such as
AlgaeBase and WoRMs. Where there was only information about ancestral or current
environment at a higher taxonomic level (i.e. at the family level but no generic or species
information) we assumed it was the same environment for the species in our dataset. We
found information on the ancestral environment of each species through broad reviews on
the origins of multicellularity including Bonner 1998, Knoll 2011 & Umen 2014 [11,21,28] . It
is important to stress that we were interested in the ancestral environment *when*
*multicellularity evolved* and therefore that was not always the same as the ancestral
environment for the whole lineage, including unicellular groups (e.g. for the Fungi, James *et*
*al.* 2006).

Of the 139 species in the dataset, 18 species had a terrestrial ancestral environment and 121
species had an aquatic ancestral environment. For the current environment, 84 species are
aquatic, 43 are terrestrial and 12 are unknown. Our full dataset ~~includes~~included 139 species
but only 121 of these species ~~where we have had~~ complete data on the number of cell types,
total number of cells, current environment, ancestral environment, and the mode of group
formation.

***Independent transitions to multicellularity***

Using information from published papers, we identified that within the eukaryotes there have
been at least 14 independent transitions to multicellularity (both facultative and obligate)
(Table 1, Figure 3a3). However, we have most likely underestimated the number of
transitions in several groups due to uncertainty about the number of independent transitions
within them. For example, it is thought that there have been at least 2 transitions to obligate
multicellularity within the Fungi [27,28] and ~~many~~multiple transitions to facultative
multicellularity in the green algae [21] and in the red algae [18]. Therefore, our analyses are
conservative and assumed just 1 transition within each group.

**Statistical Methods**

***Scaling relationships***

As a first step in analyzing the data we began with a least square regression to estimate a
and b in the scaling equation $\log_{10}y = \log_{10}a + b\log_{10}M$ meant to describe the dependence of
the number of cell types (y) on the total number of cells (M). We used ordinary least squares
(OLS) regression to evaluate the existence of allometry and estimate the intercept and slope
in the scaling of $\log_{10}(\text{cell type})$ against $\log_{10}(\text{cell number})$ across all data. We used OLS
regression rather than RMA regression even though our X variables contained measurement
error, based on the X-Y symmetry principle of Smith (2009). We note however, that OLS and
RMA approaches yielded very similar results, just with steeper slopes in all cases. We then
used the package 'segmented' in R [43] to test if there is a 'breakpoint' in the regression –
the point at which the shape of the relationship changes abruptly. We then used the OLS
approach described above to evaluate scaling relationships on either side of the breakpoint.

We also noted that the scatter plots producing average scaling relationships appeared
triangular and thus hypothesized that they reflect a constraint function such that total number
of cells is necessary, but not sufficient to explain variation in number of cell types [44,45]. To
test this hypothesis, we used least quantile regressions to describe scaling for the upper
ninetieth quantiles of the overall plot and separately for scaling relationships on either side of
the breakpoint [46,47].

~~As a first step in analyzing the data we began with a least square regression to estimate a~~
~~and b in the scaling equation $\log_{10}y = \log_{10}a + b\log_{10}M$ and describe nature of the dependence~~

Commented [RF5]: This is what we should refer to in the reviewers comments when they say that red algae is at least 2 or whatever. Say that we are trying to be conservative or something and that it won't change our results?

of the number of cell types on the total number of cells. We used the R package 'lmodel2',
we used reduced major axis (RMA) regression to estimate the intercept and slope in the
scaling of $\log_{10}(\text{cell type})$ against $\log_{10}(\text{cell number})$ across all data, for small species and for
large species. RMA is an appropriate line-fitting method in cases when measurement of both
Y and X variables are potentially associated with systematic error (e.g. the probability that
cell number was precisely measured decreased with increasing body sizes) [42]. RMA (also
known as standardized major axis regression) equally weights distances from the regression
line in both X and Y directions, with the major axis reflecting the first principal components
axis yielded by the covariance matrix, and fitted through the centroid of the data [42]. We
then used the package 'segmented' in R [43] to test if there is a 'breakpoint' in the regression
—the point at which the shape of the relationship changes dramatically. This allowed us to
estimate the different scaling relationships of small versus larger multicellular species.

We also noted that the scaling relationships appeared triangular and thus hypothesized that
they reflect a constraint function such that total number of cells is necessary, but not sufficient
to explain variation in number of cell types [44,45]. To test this hypothesis, we used least
absolute deviation regression to describe scaling for the upper ninetieth quantiles of the
overall plot and separately for the small and large taxa plots [46,47].

**Bayesian analyses**

We used the statistical package MCMCglmm [48] to run Bayesian general linear models with
Markov Chain Monte Carlo (MCMC) estimation. We fitted three models. Firstly, we tested
whether the environment affected the way in which multicellular groups form by fitting a model
with group formation as a categorical response variable and the ancestral environment as a
categorical explanatory variable (Table S3). Secondly, we tested whether the environment
affected the likelihood of obligate or facultative multicellularity by fitting a model with
obligate/facultative as a categorical response variable and the ancestral environment as a
categorical explanatory variable (Table S4).

Finally, we tested whether multicellular complexity differed between lineages living on the
land versus in aquatic environments by fitting a multi-response model with several
explanatory variables using the number of cell types and the logarithm of total number of cells
as poissonPoisson and Gaussian response variables respectively (Table S4). This allowed
878 us to use both number of cell types and the total number of cells as a combined measure of

Formatted: Font color: Custom Color(RGB(34,34,34)),
Pattern: Clear (White)

Formatted: Font color: Custom Color(RGB(34,34,34)),
Pattern: Clear (White)

Formatted: Font: 10 pt, Bold, Font color: Custom
Color(RGB(34,34,34)), Pattern: Clear (White)

multicellular complexity, rather than having to run several analyses using different response
variables. We fitted several categorical fixed effects: the current environment (aquatic or
terrestrial), whether the species is obligately or facultatively multicellular, and the mode of
group formation (non-clonal or clonal) to control for the known effects of group formation and
obligateness on multicellular complexity [4].

In the first two models, we used uninformative inverse-gamma priors because we had a
categorical response variable. We also fixed the residual variance to 1 and specified family
= categorical. In the final model, we used uninformative priors because we had a multi-
response model with both ~~poisson~~Poisson and Gaussian response variables and categorical
explanatory variables. We ran the models for 6000000 iterations, with a burn-in of 1000000
and a thinning interval of 1000. These were the values that optimised the chain length whilst
also allowing our models to converge, which we assessed visually using VCV traceplots. We
then ran each model three times and used the Gelman-Rubin diagnostic to quantitatively
check for convergence. We ~~showed~~assumed that our models had converged when the PSR
was < 1.1.

We calculated the correlations between the number of cell types and the total number of cells
~~(i.e. cov(number of cell types, total number of cells)/sqrt(var(number of cell types)* var(total
number of cells)))~~ for species in different environments. We tested if the correlation was
significantly different between environments by examining if the 95% credible interval of the
difference between the correlations spanned 0, and calculating the % of iterations where the
correlation for species living in aquatic environments was greater than that for those living on
land.

**Phylogeny construction**

We built the phylogeny for this study using the Open Tree of Life (opentreeoflife.org), which
creates synthetic trees built from published phylogenies and taxonomic information. We then
used the R package 'rotl' that interacts with the online database and constructs phylogenies
(<https://cran.r-project.org/web/packages/rotl/index.html>). For the majority of species in our
dataset, the exact species was also present in a published phylogeny and so we could use
phylogenetic information about that species. However, for a few species that were not present
in the Open Tree of Life dataset, we had to assign instead a closely related species in the
same genera or use a family-level classification. Due to the fact that most species in our

dataset represent phylogenetically distant groups on the eukaryotic tree and our phylogeny
does not include branch lengths, we ~~were~~are confident that this compromise did not affect
our statistical analysis. The phylogeny presented in Figure 3 was created for visual purposes
using Anvi (anvi-server.org).

**Funding**

RMF was supported by a Carlsberg Distinguished Post-doctoral Fellowship (CF16-0336)
hosted by JJB and JZS was supported by a European Research Council Starting Grant
(ELEVATE).

**Acknowledgements**

We thank Stuart West and Guy Cooper for thought-provoking discussions and comments
and Stefania Kapsetaki, Jordan Okie, and Jamie Gillooly for helpful edits on a final version of
the manuscript. We also thank three reviewers for their helpful comments that significantly
improved this manuscript. We dedicate this paper to the memory of John Tyler Bonner.

**References**

- 1. Bonner, J. T. 2004 Perspective: the size-complexity rule. *Evolution* **58**, 1883–1890.
 - 2. Buss, L. W. 1983 Evolution, development, and the units of selection. *Proceedings of the*
*National Academy of Sciences of the United States of America* **80**, 1387–1391.
 - 3. Herron, M. D. 2016 Origins of multicellular complexity: Volvox and the volvocine algae. *Mol.*
*Ecol.* **25**, 1213–1223. (doi:10.1111/mec.13551)
 - 4. Fisher, R. M., Cornwallis, C. K. & West, S. 2013 Group Formation, Relatedness, and the
Evolution of Multicellularity. *Current Biology* **23**, 1120–1125. (doi:10.1016/j.cub.2013.05.004)
 - 5. Strassmann, J. E., Zhu, Y. & Queller, D. C. 2000 Altruism and social cheating in the social
amoeba *Dictyostelium discoideum*. *Nature* **408**, 965–967.
 - 6. Cooper, G. A. & West, S. 2018 Division of labour and the evolution of extreme specialization.
*Nature Publishing Group* **2**, 1161–. (doi:10.1038/s41559-018-0564-9)
 - 7. Velicer, G. J., Kroos, L. & Lenski, R. E. 2000 Developmental cheating in the social bacterium
*Myxococcus xanthus*. *Nature* **404**, 598–. (doi:10.1038/35007066)
 - 8. Kuzdzal-Fick, J. J., Fox, S. A., Strassmann, J. E. & Queller, D. C. 2011 High Relatedness Is
Necessary and Sufficient to Maintain Multicellularity in *Dictyostelium*. *Science* **334**, 1548–
1551. (doi:10.1126/science.1213272)
 - 9. Regenberg, B., Hanghøj, K. E., Andersen, K. & Boomsma, J. J. 2016 Clonal yeast biofilms

- can reap competitive advantages through cell differentiation without being obligatorily
multicellular. *Proceedings of the Royal Society B: Biological Sciences* **283**, 20161303.
(doi:10.1098/rspb.2016.1303)
- 10. Bell, G. & Mooers, A. 1997 Size and complexity among multicellular organisms. *Biological*
*Journal of the Linnean Society* **60**, 345–363.
- 11. Bonner, J. T. 1998 The origins of multicellularity. *Integrative Biology Issues News and*
*Reviews* **1**, 27–36.
- 12. Peters, R. H. 1986 *The Ecological Implications of Body Size*. Cambridge University Press.
- 13. Thompson, D. W. 1961 *On Growth and Form*. Cambridge University Press.
- 14. Brown, J. H., Gillooly, J. F., Allen, A. P., Savage, V. M. & West, G. B. 2004 Toward a
metabolic theory of ecology. *Ecology*, 1771–1789.
- 15. Beardall, J., Allen, D., Bragg, J., Finkel, Z. V., Flynn, K. J., Quigg, A., Rees, T. A. V.,
Richardson, A. & Raven, J. A. 2009 Allometry and stoichiometry of unicellular, colonial and
multicellular phytoplankton. *New Phytol.* **181**, 295–309. (doi:10.1111/j.1469-
8137.2008.02660.x)
- 16. Tice, A. K., Silberman, J. D., Walthall, A. C., Le, K. N. D., Spiegel, F. W. & Brown, M. W.
2016 Sorodiphrys stercorea: Another Novel Lineage of Sorocarpic Multicellularity. *Journal*
*of Eukaryotic Microbiology* **63**, 623–628. (doi:10.1111/jeu.12311)
- 17. Brown, M. W., Kolisko, M., Silberman, J. D. & Roger, A. J. 2012 Aggregative Multicellularity
Evolved Independently in the Eukaryotic Supergroup Rhizaria. *Current Biology* **22**, 1123–
1127. (doi:10.1016/j.cub.2012.04.021)
- 18. Knoll, A. 2011 The Multiple Origins of Multicellularity. *Annual Review of Earth and Planetary*
*Sciences* **39**, 217–239.
- 19. Cock, J. M. et al. 2010 The Ectocarpus genome and the independent evolution of
multicellularity in brown algae. *Nature* **465**, 617–621. (doi:10.1038/nature09016)
- 20. Schaap, P. et al. 2006 Molecular phylogeny and evolution of morphology in the social
amoebas. *Science* **314**, 661–663. (doi:10.1126/science.1130670)
- 21. Umen, J. G. 2014 Green Algae and the Origins of Multicellularity in the Plant Kingdom. *Cold*
*Spring Harb Perspect Biol* **6**. (doi:10.1101/cshperspect.a016170)
- 22. King, N. 2004 The unicellular ancestry of animal development. *Dev Cell* **7**, 313–325.
- 23. Sugimoto, H. & Endoh, H. 2006 Analysis of fruiting body development in the aggregative
ciliate *Sorogena stoianovitchae* (Ciliophora, Colpodea). *Journal of Eukaryotic Microbiology*
**53**, 96–102. (doi:10.1111/j.1550-7408.2005.00077.x)
- 24. Olive, L. S. & Blanton, R. L. 1980 Aerial Sorocarp Development by the Aggregative Ciliate,
*Sorogena stoianovitchae*. *Protozoology* **27**, 293–299.
- 25. Summers, F. 1938 Some aspects of normal development in the colonial ciliate *Zoothamnion*
*alternans*. *Biol. Bull.*, 117–129.
- 26. Brown, M. W., Spiegel, F. W. & Silberman, J. D. 2009 Phylogeny of the ‘forgotten’ cellular
slime mold, *Fonticula alba*, reveals a key evolutionary branch within Opisthokonta. *Mol Biol*

- *Evol* **26**, 2699–2709. (doi:10.1093/molbev/msp185)
- 27. Nguyen, T. A., Cisse, O. H., Wong, J. Y., Zheng, P., Hewitt, D., Nowrousian, M., Stajich, J. E.
& Jedd, G. 2017 Innovation and constraint leading to complex multicellularity in the
Ascomycota. *Nat Commun* **8**. (doi:10.1038/ncomms14444)
- 28. Knoll, A. H. 2011 The Multiple Origins of Complex Multicellularity. *Annu Rev Earth Pl Sc* **39**,
217–239. (doi:10.1146/annurev.earth.031208.100209)
- 29. Hesse, M. & Kusel-Fetzmann, E. 1989 Life cycle and ultrastructure of *Ducellieria chodati*
(Oomycetes). *Plant systematics and evolution* **165**, 1–15.
- 30. Everhart, S. & Keller, H. 2008 Life history strategies of corticolous myxomycetes: the life
cycle, plasmodial types, fruiting bodies, and taxonomic orders. *Fungal Diversity* **29**, 1–16.
- 31. James, T. Y. et al. 2006 Reconstructing the early evolution of Fungi using a six-gene
phylogeny. *Nature* **443**, 818–822. (doi:10.1038/nature05110)
- 32. Brückner, S. & Mösch, H.-U. 2011 Choosing the right lifestyle: adhesion and development in
*Saccharomyces cerevisiae*. *FEMS Microbiology Reviews* **36**, no–no. (doi:10.1111/j.1574-
6976.2011.00275.x)
- 33. Fisher, R. M. & Regenberg, B. 2019 Multicellular group formation in *Saccharomyces*
*cerevisiae*. *Proceedings of the Royal Society B: Biological Sciences* **286**, 20191098.
(doi:10.1098/rspb.2019.1098)
- 34. Grosberg, R. K. & Strathmann, R. R. 2007 The Evolution of Multicellularity: A Minor Major
Transition? *Annu Rev Ecol Evol S* **38**, 621–654.
(doi:10.1146/annurev.ecolsys.36.102403.114735)
- 35. West, S., Fisher, R. M., Gardner, A. & Kiers, E. T. 2015 Major evolutionary transitions in
individuality. *Proceedings of the National Academy of Sciences of the United States of*
*America* (doi:10.1073/pnas.1421402112)
- 36. Boomsma, J. J. 2007 Kin selection versus sexual selection: why the ends do not meet.
*Current Biology* **17**, 673–683.
- 37. Boomsma, J. J. 2009 Lifetime monogamy and the evolution of eusociality. pp. 3191–
3207. (doi:10.1098/rstb.2009.0101)
- 38. Boomsma, J. J. & Gawne, R. 2018 Superorganismality and caste differentiation as points of
no return: how the major evolutionary transitions were lost in translation. *Biol Rev Camb*
*Philos Soc* **93**, 28–54. (doi:10.1111/brv.12330)
- 39. Fisher, R. M., Henry, L. M., Cornwallis, C. K., Kiers, E. T. & West, S. 2017 The evolution of
host-symbiont dependence. *Nat Commun* **8**, 1–8. (doi:10.1038/ncomms15973)
- 40. Lucky, A., Trautwein, M. D., Guénard, B. S., Weiser, M. D. & Dunn, R. R. 2013 Tracing the
rise of ants - out of the ground. *PLoS ONE* **8**, e84012. (doi:10.1371/journal.pone.0084012)
- 41. Arendt, D. et al. 2016 The origin and evolution of cell types. *Nature Publishing Group*, 1–15.
(doi:10.1038/nrg.2016.127)
- 42. Warton, D. I., Wright, I. J., Falster, D. S. & Westoby, M. 2006 Bivariate line-fitting methods for
allometry. *Biol Rev Camb Philos Soc* **81**, 259–291. (doi:10.1017/S1464793106007007)

- 43. Muggeo, V. M. R. 2003 Estimating regression models with unknown break-points. *Stat Med*
**22**, 3055–3071. (doi:10.1002/sim.1545)
- 44. Cade, B. S. & Noon, B. R. 2003 A gentle introduction to quantile regression for ecologists.
*Frontiers in Ecology and the Environment* **1**, 412–420.
- 45. Brown, J. H. 1995 *Macroecology*. University of Chicago Press.
- 46. Koenker, R. 2005 *Quantile Regression*. Cambridge University Press.
- 47. Cade, B. & Richards, J. 2013 *User Manual for Blossom Statistical Software*. CreateSpace.
- 48. Hadfield, J. D. & Nakagawa, S. 2010 General quantitative genetic methods for comparative
biology: phylogenies, taxonomies and multi-trait models for continuous and categorical
characters. *Journal of Evolutionary Biology* **23**, 494–508. (doi:10.1111/j.1420-
9101.2009.01915.x)

Formatted: Font: Not Bold

Supplementary Information and Figures

Figure S1: Phylogeny of the multicellular lineages in our dataset. Phylogeny created using the Open Tree of Life and 'rotl' package in R and edited using the online software Interactive Tree of Life. Each lineage that independently evolved multicellularity within the Eukaryotes is highlighted in a different colour and the species that appear in our dataset are given at the tips.

**Supplementary Statistical Results Tables**

**Table S1:** Statistical results from ordinary least squares regression (OLS) on the full dataset (all), on
 the lower side of the breakpoint (small species) and the higher side of the breakpoint (large species).
 *slopes are significantly different from 0

**Table S1:** Statistical results from Least squares (OLS) and Reduced Major Axis (RMA) Regression
 on the full dataset, small species and large species. *slopes are significantly different from 0

Model	N	Intercept (2.5% - 97.5% confidence intervals)	Slope (2.5% - 97.5% confidence intervals)	R ²	p value
OLS (all)	126	0.03 (-0.09 – 0.15)	0.11 (0.10 – 0.13)	0.64	0.01*
OLS (small)	50	-0.23 (-0.37 – 0.09)	0.21 (0.16 – 0.26)	0.61	0.01*
OLS (large)	76	0.46 (0.08 – 0.83)	0.07 (0.03 – 0.11)	0.15	0.01*
RMA (all)	126	-0.16 (-0.27 – -0.06)	0.14 (0.13 – 0.16)	N/A	N/A
RMA (small)	50	-0.38 (-0.52 – -0.27)	0.27 (0.23 – 0.33)	N/A	N/A
RMA (large)	76	-0.59 (-1.00 – -0.26)	0.18 (0.15 – 0.23)	N/A	N/A

**Table S2 :** Statistical results from the 90% ~~Regression~~ regression on the small and large species data.

Model	N	Intercept (2.5% - 97.5% confidence intervals)	Slope (2.5% - 97.5% confidence intervals)
90% Quantile regression (small)	50	-0.08 (-0.50 – 0.35)	0.25 (0.10 – 0.40)
90% Quantile regression (large)	76	0.44 (0.13 – 0.76)	0.13 (0.10 – 0.17)

**Table S3:** Analysis of the effect of the ancestral environment on the mode of multicellular group
 formation, taking into account phylogenetic relationships using MCMCglmm.

Response: Mode of group formation				
Fixed effect		N	Posterior mode (CI)	pMCMC
	Aquatic	121	4.80 (1.26 – 7.58)	

Ancestral environment	Terrestrial	18	-1.43 (-4.78 – 0.80)	
	Difference		5.74 (2.91 – 9.79)	0.0008

**Table S4:** Analysis of the effect of the ancestral environment on whether a species is obligately or
facultatively multicellular, taking into account phylogenetic relationships using MCMCglmm. N total =
139.

Response: Obligate or facultative

Fixed effect		N	Posterior mode (CI)	pMCMC
Ancestral environment	Aquatic	121	4.04 (2.39 – 5.97)	
	Terrestrial	18	-2.02 (-4.09 - -0.49)	
	Difference		6.59 (4.29 – 8.72)	< 0.0001

**Table S5:** Analysis of the effect of the current environment, whether the species is obligately or
facultatively multicellular and the mode of group formation on the number of cell types and the total
number of cells, taking into account phylogenetic relationships using MCMCglmm. N total = 121 as
any missing values were removed ~~for~~before full analyses.

Response: The number of cell types

Fixed Effect		N	Posterior mode (CI)	pMCMC
Current environment	Aquatic	85	-0.11 (-1.56 – 1.70)	
	Terrestrial	36	0.53 (-0.72 – 2.36)	
	Difference		-0.77 (-1.42 - -0.11)	0.02
Obligate or facultative	Obligate	114	1.97 (-0.36 – 3.87)	
	Facultative	7	0.38 (-1.49 – 1.77)	
	Difference		1.68 (0.06 – 3.24)	0.02
Mode of group formation	Clonal	116	0.11 (-1.75 – 2.33)	
	Non-clonal	5	0.18 (-1.46 – 1.82)	
	Difference		-0.19 (-1.66 – 1.43)	0.43

Response: Total number of cells

Fixed Effect		N	Posterior mode	pMCMC
Current environment	Aquatic	85	9.80 (-5.03 – 20.82)	
	Terrestrial	36	13.21 (-2.16 – 23.22)	
	Difference		-2.79 (-9.04 – 1.81)	0.12
Obligate or facultative	Obligate	114	7.95 (0.94 – 12.97)	
	Facultative	7	3.40 (-1.66 – 9.01)	
	Difference		3.89 (0.65 – 6.62)	0.007
Mode of group formation	Clonal	116	1.78 (-3.36 – 7.71)	
	Non-clonal	5	3.23 (-1.83 – 9.15)	
	Difference		1.46 (-1.98 – 4.51)	0.23

Correlations

Number of cell types : Total number of cells	Phylogenetic correlation	0.90 (0.72 – 0.96)	< 0.0001
	Phenotypic correlation	0.56 (0.19 – 0.76)	0.004

**Table S6:** The full dataset, with corresponding references, for all of the data used in our analyses.

Species	Obligate or facultative	Number of cell types	Total number of cells	Group formation	Current environment	Ancestral environment	Reference(s)
Acrasis rosea	facultative	2	707.9457844	non-clonal	terrestrial	terrestrial	Raper 1984, Bonner 2003, Brown et al. 2011
Acytostelium	facultative	1	NA	non-clonal	terrestrial	terrestrial	Bonner 2003, Bourke 2011
Aelosoma tenebrarum	obligate	12	50118.72	clonal	NA	aquatic	Brace 1901
Alaria marginata	obligate	14	1.00E+12	clonal	aquatic	aquatic	Kain 1979, Charrier et al. 2007
Anthoceros himalayensis	obligate	12	39810.71706	clonal	terrestrial	aquatic	Henra & Handoo 1953, Nishiyama 2007
Arabidopsis thaliana	obligate	30		clonal	terrestrial	aquatic	Carroll 2001, Lin et al. 1999
Ascophyllum nodosum	obligate	6	6.31E+11	clonal	aquatic	aquatic	Rawlence 1978, Charrier et al. 2007
Asperococcus fistulosus	obligate	5	10000000000	clonal	aquatic	aquatic	Bold & Wynne 1978, Charrier et al. 2007
Astrephomene gubernaculifera	obligate	2	64	clonal	aquatic	aquatic	Stein 1958, Herron & Michod 2008, Hallmann 2011
Astrephomene perforata	obligate	2	64	clonal	aquatic	aquatic	Herron & Michod 2008, Hallmann 2011
Basichlamys sacculifera	obligate	1	4	clonal	aquatic	aquatic	Nozaki et al. 1996, Hallmann 2011
Beckerlla scalaramosa	obligate	12	31622776600	clonal	aquatic	aquatic	Kraft 1976, Graham 1985
Botryocladia wynnei	obligate	6	3981071.706	clonal	aquatic	aquatic	Ballantine 1985, Graham 1985
Callinectes sapidus	obligate	69	3.16E+11	clonal	NA	aquatic	Johnson 1980
Candida albicans	facultative	3	NA	clonal	NA	terrestrial	Engelberg et al. 1998, Whiteway & Bachewich 2007
Canis familiaris	obligate	99	5.01E+13	clonal	terrestrial	aquatic	Adam et al. 1983
Carpomitra cabrecae	obligate	7	2511886432	clonal	aquatic	aquatic	Motomura et al. 1985, Charrier et al. 2007
Chlamydomonas reinhardtii	facultative	1	71	clonal	aquatic	aquatic	Bold & Wynne 1985, Kirk 1998, Lurling et al. 2006, Becks et al. 2010

Chordaria flagelliformis	obligate	6	3981071706	clonal	aquatic	aquatic	Kornmann 1962, Charrier et al. 2007
Chordaria linearis	obligate	6	10000000000	clonal	aquatic	aquatic	Searles 1980, Charrier et al. 2007
Cladostephus verticillatus	obligate	8	125892541.2	clonal	aquatic	aquatic	Sauvageau 1907, Charrier et al. 2007
Colpomenia sinuosa	obligate	5	1995262315	clonal	aquatic	aquatic	Wynne 1972, Charrier et al. 2007
Conocephalum conicum	obligate	15	3162277.66	clonal	terrestrial	aquatic	Maybrook 1914, Nishiyama 2007
Croomia pauciflora	obligate	42	25118864320	clonal	terrestrial	aquatic	Tomlinson & Ayensu 1968
Cutleria	obligate	7	3162277660	clonal	aquatic	aquatic	Bold & Wynne 1978, Charrier et al. 2007
Cyanea cyanea	obligate	22	1.00E+13	clonal	aquatic	aquatic	Hyman 1940
Cyathodium barodae	obligate	13	50118723.36	clonal	terrestrial	aquatic	Chavran 1937, Nishiyama 2007
Dasybranchus caducus	obligate	10	31622.7766	clonal	NA	aquatic	Bookhaut 1957
Desmarestia antarctica	obligate	7	6.31E+11	clonal	aquatic	aquatic	Moe & Silva 1989, Charrier et al. 2007
Dictyosiphon hirsutus	obligate	6	39810717060	clonal	aquatic	aquatic	Peters 1992, Charrier et al. 2007
Dictyostelium discoideum	facultative	3	100000	non-clonal	terrestrial	terrestrial	Kaiser 1986, Raper 1940, Rokas 2008
Dictyostelium fasciculatum	facultative	2	NA	non-clonal	terrestrial	terrestrial	
Dictyostelium minutum	facultative	2	3162.27766	non-clonal	terrestrial	terrestrial	Kaiser 198
Dictyostelium purpureum	facultative	2	NA	non-clonal	terrestrial	terrestrial	Mehdiabadi et al. (2009)
Dictyota binghamiae	obligate	4	2.51E+11	clonal	aquatic	aquatic	Foster et al. 1972, Charrier et al. 2007
Diurodrilus westheidi	obligate	14	3548.134	clonal	NA	aquatic	Kristensen & Niilon 1982
Ducellieria chodati	obligate	1	40	clonal	aquatic	aquatic	Hesse et al. 1989
Durvillaea antarctica	obligate	6	1.00E+12	clonal	aquatic	aquatic	Naylor 1949, Charrier et al. 2007
Ectocarpus siliculosus	obligate	4	316227.766	clonal	aquatic	aquatic	Knight 1931, Charrier et al. 2007
Elachista fucicola	obligate	5	15848931.92	clonal	aquatic	aquatic	Koeman & Cortel-Breeman 1976, Charrier et al. 2007

Eudorina cylindrica	obligate	2	16	clonal	aquatic	aquatic	Herron & Michod 2008, Hallmann 2011
Eudorina elegans	obligate	1	32	clonal	aquatic	aquatic	Hallmann 2011
Eudorina minodii	obligate	1	32	clonal	aquatic	aquatic	Hallmann 2011
Eudorina unicocca	obligate	1	32	clonal	aquatic	aquatic	Yamada et al. 2008, Hallmann 2011
Farlowia mollis	obligate	7	3162277660	clonal	aquatic	aquatic	Abbott 1962, Graham 1985
Fonticula alba	facultative	2		non-clonal	terrestrial	terrestrial	Brown et al. 2009
Fucus vesiculosus	obligate	7	3.16E+12	clonal	aquatic	aquatic	McCully 1966, Charrier et al. 2007
Fuirena ciliaris	obligate	44	25118864320	clonal	terrestrial	aquatic	Govindarajalu 1969, Graham 1985
Funaria hygrometrica	obligate	20	251188643.2	clonal	terrestrial	aquatic	Puri 1981, Nishiyama 2007
Gloeophycus koreanum	obligate	12	63095734450	clonal	aquatic	aquatic	Lee & Yoo 1979, Graham 1985
Gonium multicocum	obligate	1	32	clonal	aquatic	aquatic	Hallmann 2011
Gonium octonarium	obligate	1	32	clonal	aquatic	aquatic	Hallmann 2011
Gonium pectorale	obligate	1	16	clonal	aquatic	aquatic	Hallmann 2011, Herron & Michod 2008, Stein 1959
Gonium quadratum	obligate	1	16	clonal	aquatic	aquatic	Hallmann 2011
Gonium viridistellatum	obligate	1	16	clonal	aquatic	aquatic	Hallmann 2011
Gymnoascus reessii	obligate	5	15848.93192	clonal	terrestrial	terrestrial	Gaetano 1986, Graham 1985
Halymenia asymmetrica	obligate	13	63095734450	clonal	aquatic	aquatic	Gaetano 1986, Graham 1985
Haplospora globosa	obligate	4	25118864320	clonal	aquatic	aquatic	Kuhlenkamp & Muller 1985, Charrier et al. 2007
Helminthostachys zeylandica	obligate	5	70794578.44	clonal	terrestrial	aquatic	Lang 1902, Graham 1985
Heteroralsia saxicola	obligate	9	794328234.7	clonal	aquatic	aquatic	Kawai 1989, Charrier et al. 2007
Himantothallus grandifolius	obligate	14	1.58E+12	clonal	aquatic	aquatic	Wiencke & Clayton 1990, Charrier et al. 2007
Hirudo medicinalis	obligate	26	19952623150	clonal	NA	aquatic	Mann 1962
Homo sapiens	obligate	200	1.00E+14	clonal	terrestrial	aquatic	Valentine et al. 1994

Hummia onusta	obligate	5	100000000	clonal	aquatic	aquatic	Fiore 1977, Charrier et al. 2007
Hydra attenuata	obligate	15	63095.73445	clonal	aquatic	aquatic	Campbell & Bode 1983, Glauber et al. 2010
Hymenophyllum tunbridgensis	obligate	15	7079457844	clonal	terrestrial	aquatic	Boodle 1900, Graham 1985
Isthmoploea sphaerophora	obligate	3	15848.932	clonal	aquatic	aquatic	Rueness 1974, Charrier et al. 2007
Kurogiella saxatilis	obligate	7	25118864320	clonal	aquatic	aquatic	Kawai 1993, Charrier et al. 2007
Laminaria dentigera	obligate	14	1.26E+11	clonal	aquatic	aquatic	Kain 1979, Charrier et al. 2007
Leathesia difformis	obligate	6	39810717060	clonal	aquatic	aquatic	Bold & Wynne 1978, Charrier et al. 2007
Lemna minor	obligate	18	794328.2347	clonal	terrestrial	aquatic	Daubs 1965, Graham 1985
Lomandra hermaphroditicum	obligate	36	35481338920	clonal	terrestrial	aquatic	Fahn 1954, Graham 1985
Lumbricus terrestris	obligate	57	10000000000	clonal	terrestrial	aquatic	Stephenson 1930
Mamillaria elongata	obligate	27	63095734450	clonal	terrestrial	aquatic	Darbishire 1904
Membranoptera subtropica	obligate	12	6309573.445	clonal	aquatic	aquatic	Schneider & Eisemann 1979, Graham 1985
Monoclea forsteri	obligate	13	3162277.66	clonal	terrestrial	aquatic	Shuster 1984, Nishiyama 2007
Morone saxatilis	obligate	122	2.51E+11	clonal	aquatic	aquatic	Groman 1982
Mus musculus	obligate	102	2.00E+11	clonal	terrestrial	aquatic	Gude et al. 1982
Nais variabilis	obligate	13	251188.6432	clonal	NA	aquatic	Stephenson 1908
Neodilsea natashae	obligate	12	19952623150	clonal	aquatic	aquatic	Linstrom 1984, Graham 1985
Ophioglossum palmatum	obligate	14	6309573445	clonal	terrestrial	aquatic	Chrysler 1941, Graham 1985
Pandorina colemaniae	obligate	1	16	clonal	aquatic	aquatic	Bold & Wynne 1985, Hallmann 2011
Pandorina morum	obligate	1	16	clonal	aquatic	aquatic	Bold & Wynne 1985, Hallmann 2011
Periplaneta americana	obligate	50	3162277660	clonal	terrestrial	aquatic	Smith 1968
Petermannia cirrhosa	obligate	39	25118864320	clonal	terrestrial	aquatic	Tomlinson & Ayensu 1968, Graham 1985
Physarum polycephalum	facultative	2	300	clonal	terrestrial	terrestrial	Stephenson & Stempen 1994, Baldauf &

							Doolittle 1996, Everhart et al. 2008
Pinus monophylla	obligate	30	10000000000	clonal	terrestrial	aquatic	Foster & Gifford 1974
Pisione remota	obligate	11	44668.35922	clonal	NA	aquatic	Akesson 1961
Platydorina caudata	obligate	1	32	clonal	aquatic	aquatic	Hallmann 2011
Pleodorina californica	obligate	2	128	clonal	aquatic	aquatic	Hallmann 2011
Pleodorina illinoisensis	obligate	2	32	clonal	aquatic	aquatic	Bonner 2004, Hallmann 2011
Pleodorina indica	obligate	2	128	clonal	aquatic	aquatic	Bonner 2004, Hallmann 2011
Pleodorina japonica	obligate	2	128	clonal	aquatic	aquatic	Hallmann 2011
Pleurobrachia	obligate	13	10000	clonal	aquatic	aquatic	Hyman 1940
Pocheina flagellata	facultative	2	NA	non-clonal	terrestrial	terrestrial	Bell & Mooers 1997, Brown et al. 2011
Pocheina rosea	facultative	2	NA	non-clonal	terrestrial	terrestrial	Bell & Mooers 1997, Brown et al. 2011
Pogonatum stevensii	obligate	21	707945784.4	clonal	terrestrial	aquatic	Chopra & Sharna 1958, Nishiyama 2007
Polysphondylium pallidum	facultative	2	NA	non-clonal	terrestrial	terrestrial	Stenhouse & Williams 1980
Polysphondylium violaceum	facultative	2	NA	non-clonal	terrestrial	terrestrial	Bonner 1959
Polytrichum commune	obligate	26	1000000000	clonal	terrestrial	aquatic	Puri 1981, Nishiyama 2007
Pomatoceros triquetter	obligate	12	70794.578	clonal	NA	aquatic	Segrove 1941
Ralfsia verrucosa	obligate	8	630957344.5	clonal	aquatic	aquatic	Loiseaux 1968, Charrier et al. 2007
Saccharomyces cerevisiae	facultative	3	NA	clonal	NA	terrestrial	Ratcliff et al. 2011, Koschwanez et al. 2011
Sagittaria lancifolia	obligate	42	1.00E+11	clonal	terrestrial	aquatic	Stant 1964, Graham 1985
Salmo gairdneri	obligate	116	2.51E+11	clonal	aquatic	aquatic	Yasutake 1983
Salpingoeca rosetta	facultative	NA	29	clonal	aquatic	aquatic	Fairclough et al. 2010, Dayel et al. 2011
Sarconema scinaoides	obligate	13	2511886432	clonal	aquatic	aquatic	Papenfuss & Edelstein 1974, Graham 1985

Schimitzia hiscockiana	obligate	14	2.00E+11	clonal	aquatic	aquatic	Maggs & Guiry 1985, Graham 1985
Schimmelmannia dawsonii	obligate	11	2.51E+11	clonal	aquatic	aquatic	Acleto 1972, Graham 1985
Scytosiphon lomentria	obligate	4	794328234.7	clonal	aquatic	aquatic	Clayton 1976, Charrier et al. 2007
Selenipedium palmifolium	obligate	35	12589254120	clonal	terrestrial	aquatic	Rosso 1966, Graham 1985
Sorogena stoianovitchae	facultative	1	500	non-clonal	aquatic	terrestrial	Sugimoto & Endoh 2006, Blanton & Olive 1982, Olive & Blanton 1980
Sphacelaria bipinnata	obligate	9	1258925412	clonal	aquatic	aquatic	Clint 1927, Charrier et al. 2007
Sphaerobolus stellatus	obligate	9	1258925.412	clonal	terrestrial	terrestrial	Buller 1933, Knoll 2011
Sphagnum recurvum	obligate	11	891250938.1	clonal	terrestrial	aquatic	Puri 1981, Nishiyama 2007
Symphyogyna brogniarti	obligate	13	446683.5922	clonal	terrestrial	aquatic	Puri 1981, Nishiyama 2007
Syngoderma phinneyi	obligate	6	199526.2315	clonal	aquatic	aquatic	Henry & Muller 1983, Charrier et al. 2007
Tetrabaena socialis	obligate	1	4	clonal	aquatic	aquatic	Stein 1959, Herron & Michod 2008, Hallmann 2011
Volvox africanus	obligate	2	8192	clonal	aquatic	aquatic	Herron & Michod 2008, Nozaki 2011, Hallmann 2011
Volvox aureus	obligate	2	2048	clonal	aquatic	aquatic	Hallmann 2011
Volvox carteri	obligate	2	2048	clonal	aquatic	aquatic	Nishii & Miller 2010, Bonner 1998, Kaiser 2001, Kirk 1999, Hallmann 2011
Volvox dissipatrix	obligate	2	16384	clonal	aquatic	aquatic	Herron & Michod 2008, Hallmann 2011, Herron 2014
Volvox gigas	obligate	2	4096	clonal	aquatic	aquatic	Herron & Michod 2008, Hallmann 2011, Herron 2014
Volvox globator	obligate	2	16384	clonal	aquatic	aquatic	Herron & Michod 2008, Hallmann 2011, Herron 2014

Volvox obversus	obligate	2	2048	clonal	aquatic	aquatic	Herron & Michod 2008, Hallmann 2011, Herron 2014
Volvox rousseletii	obligate	2	32768	clonal	aquatic	aquatic	Herron & Michod 2008, Hallmann 2011, Herron 2014
Volvox tertius	obligate	2	1024	clonal	aquatic	aquatic	Herron & Michod 2008, Hallmann 2011, Herron 2014
Volvulina boldii	obligate	1	16	clonal	aquatic	aquatic	Herron & Michod 2008, Hallmann 2011, Herron 2014
Volvulina compacta	obligate	1	16	clonal	aquatic	aquatic	Herron & Michod 2008, Hallmann 2011, Herron 2014
Volvulina pringsheimii	obligate	1	16	clonal	aquatic	aquatic	Herron & Michod 2008, Hallmann 2011, Herron 2014
Volvulina steinii	obligate	1	16	clonal	aquatic	aquatic	Stein 1958, Hallmann 2011
Wolffia arrhiza	obligate	5	10000	clonal	terrestrial	aquatic	Luandolt 1986, Graham 1985
Wolffia microscopica	obligate	7	70794.57844	clonal	terrestrial	aquatic	Maheshwari 1954, Graham 1985
Yamadaella cenomyce	obligate	7	3981071706	clonal	aquatic	aquatic	Abbott 1970, Graham 1985
Yamadaphycus carnosa	obligate	11	1000000000	clonal	aquatic	aquatic	Mikami 1973, Graham 1985
Yamagishiella unicocca	obligate	1	32	clonal	aquatic	aquatic	Yamada et al. 2008, Herron & Michod 2008, Hallmann 2011
Zeacarpa leiomorpha	obligate	8	7943282347	clonal	aquatic	aquatic	Anderson et al. 1988, Charrier et al. 2007
Zoothamnium alterans	obligate	4	141.2537545	clonal	aquatic	aquatic	Summers 1938, Faure-Fremiet 1930

Appendix B

**Response to Reviewers comments**

**Associate Editor:**

**Board Member: 1**

**Comments to Author:**

Congratulations! We are pleased to accept your manuscript with minor revisions. This is an
interesting and well written paper that uses a phylogenetic context captures large scale
trends in evolution of multicellularity.

In your revision please address the comments of reviewer 2 with respect to classification of
cell types, as well as reviewers 1's comment on prokaryotes.

As well, before publication we also require data and code deposition for reproducibility.
While the authors link to a well documented repository containing the data published in a
previous analysis, and the updated data is provided in the supplement, the data deposition
guidelines are: "To allow others to verify and build on the work published in Royal Society
journals, it is a condition of publication that authors make available the data, code and
research materials supporting the results in the article." In order to make the work from this
publication reproducible, their expanded categorized data set, inferred phylogeny, and
associated analysis code should be shared in a dryad repository for this paper.

*The data, phylogeny and original code are now available on Dryad and we have added the*
*link and doi to the Methods section (lines 285 – 286).*

Some citations are missing in the methods section. The ROTL package was described in
Michonneau et al 2016 <https://doi.org/10.1111/2041-210X.12593>, and the OpenTree of Life
project lists appropriate citations at [https://tree.opentreeoflife.org/about/open-tree-of-](https://tree.opentreeoflife.org/about/open-tree-of-life)
[life](https://tree.opentreeoflife.org/about/open-tree-of-life) (citations to the taxonomy, datastore, and synthesis would be appropriate).

*We have added these missing citations.*

**Minor figure comments:**

The resizing of the axes between panel a and b in figure 1 makes the match between them
harder to see. Also, consider using different symbols in addition to different colors in panel
b and Figure 4. to make the figure more comprehensible in B/W and for people with limited
color vision.

*We have edited the figures so they are more 'in line' and comparable. We accept the*
*comment regarding accessibility with Figure 1b, however have decided to leave the colour*
*as is, because 14 different shapes on a small is likely to be harder to read and lessen the*
*Figures impact.*

Figure 3. Could make use of the extra space in the key to provide more visual clarification
about the bar charts, and label the axis values for the bar charts.

*We have added the cell types values to the legend, as this would be too small to add to the*
*figure and still be legible (lines 449 – 450).*

Consider B/W printing and color challenges in phylogeny figure as well - e.g. dividing line
between ciliates and red algae is hard to see even with color.

We have changed the colour of the ciliates so that there is a better distinction between the phylogenetic groups.

Reviewer(s)' Comments to Author:

Referee: 1

Comments to the Author(s)

The authors adequately addressed the referees' concerns in their revision. In particular, they clarified the statistical methods used, which answered my main concerns. As such, I believe the manuscript is suitable for publication in the current form.

As a note, in line 265 the authors refer to "prokaryotes and archaea". I think they mean "bacteria and archaea" (or otherwise simply "prokaryotes") since archaea are prokaryotes also.

We thank the reviewer for spotting this error. We have changed this to 'prokaryotes'.

Referee: 3

Comments to the Author(s)

The authors have substantially revised the manuscript. It seems that the statistical analyses reveal a trend that is inherent to the data and not imposed by the analyses, which was a major concern raised in the first round of reviews. However, we still feel that some of the problems associated with the classification of cell types in the dataset were not sufficiently addressed. In particular, we would like to see a bit more discussion on how the different authors of the datasets used in this paper classified cell types across different multicellular lineages, and how this affects the results in this paper. This is important because the classification in itself could have implicit biases in the number of cell types counted, especially because a substantial portion of the data is now fairly old (i.e., Bell 1997), and modern molecular techniques have changed our views of what counts as a cell type. Otherwise, the paper is nice!

We thank the reviewer for their comments. We have now added a section to the Methods (lines 297 – 303) where we go into more detail about how cell types have been defined. We also acknowledge this in the Discussion section (lines 229 – 232).